# How Did the Model Change? Efficiently Assessing Machine Learning API Shifts

**Lingjiao Chen, Matei Zaharia, James Y. Zou**
Stanford University
{lingjiao,jamesz}@stanford.edu matei@cs.stanford.edu

## Abstract

ML prediction APIs from providers like Amazon and Google have made it simple to use ML in applications. A challenge for users is that such APIs continuously change over time as the providers update models, and changes can happen silently without users knowing. It is thus important to monitor when and how much the ML APIs' performance shifts. To provide detailed change assessment, we model ML API shifts as confusion matrix differences, and propose a principled algorithmic framework, MASA, to provably assess these shifts efficiently given a sample budget constraint. MASA employs an upper-confidence bound based approach to adaptively determine on which data point to query the ML API to estimate shifts. Empirically, we observe significant ML API shifts from 2020 to 2021 among 12 out of 36 applications using commercial APIs from Google, Microsoft, Amazon, and other providers. These real-world shifts include both improvements and reductions in accuracy. Extensive experiments show that MASA can estimate such API shifts more accurately than standard approaches given the same budget.

## 1 Introduction

Machine learning (ML) prediction APIs have made it dramatically easier to build ML applications. For example, one can use Microsoft text API (Mic, a) to determine the polarity of a text review written by a customer, or Google speech API (Goo, b) to recognize users' spoken commands received by a smart home device. These APIs have been gaining in popularity (MLa; Chen et al., 2020), as they eliminate the need for ML users to train their own models.

Monitoring and assessing the performance of these third-party ML APIs over time, however, is under-explored. ML API providers continuously collect new data or change their model architectures (Qi et al., 2020) to update their services, which could silently help or harm downstream applications' performance. For example, as shown in Figure 1 (a) and (b), we observe a 7% overall accuracy drop of IBM speech API on the AUDIOMNST dataset in March 2021 compared March 2020. In our systematic study of 36 API and dataset combinations, there are 12 cases where the API's performance changed by more than $1\%$ on the same dataset from 2020 to 2021 (sometimes for the worse). Such performance shifts are of serious concern not only because of potential disruptions to downstream tasks, but also because consistency is often required for audits and oversight. Therefore, it is important to precisely assess shifts in an API's predictions over time. Moreover, in this assessment, it is often more informative to quantify how the entire confusion matrix of the API has changed rather than just the overall accuracy. In the IBM case in Figure 1, it is interesting that a major culprit of the drop in performance is the 2021 model mistaking "four" for "five". In other settings, changes in the confusion matrix could still cause issues even if the overall accuracy stays the same.

In this paper, we formalize the problem of assessing API shifts as estimating changes in the confusion matrix on the same dataset in a sample-efficient manner (i.e., with few calls to the API itself to minimize dollar cost). The straightforward approach is to compare the API's prediction on randomly sampled data. However, this can require a large number of API calls to estimate the confusion matrix, which is expensive given that each call costs money. To help address this challenge, we propose MASA, a principled algorithm for ML API shift assessments. MASA efficiently estimates shifts in the API's confusion matrix by clustering the dataset and adaptively sampling data from different clusters to query the API. MASA automates its sampling rate from different data clusters based on

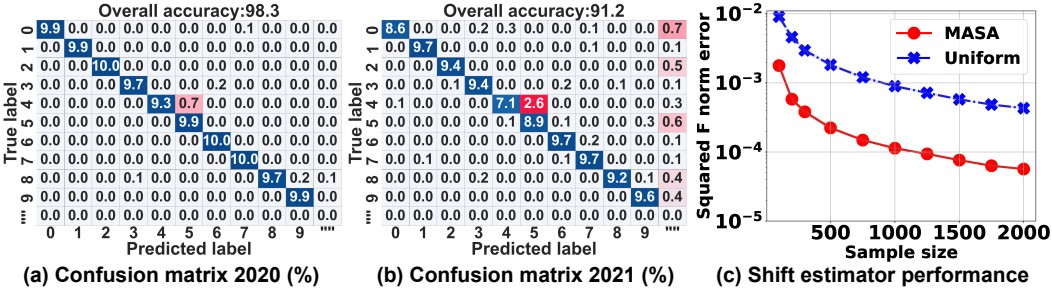

Figure 1: ML API shift for IBM speech recognition API on AMNIST, a spoken digit dataset. (a) and (b) give its (normalized) confusion matrix in April 2020 and 2021, respectively. There is an overall 7% accuracy drop. One factor is the 2021 model incorrectly predicting more "four" as "five". (c) Given a sample budget, the proposed MASA can assess the API shift with much smaller error in Frobenius norm compared to standard uniform sampling.

the uncertainty in the confusion matrix estimation. For example, MASA may query the ML API on more samples with the true label "four" than "one", if it is less sure about the estimated performance change on the former. Employing an upper-confidence-bound approach to estimate the uncertainties, MASA enjoys a low computation and space cost as well as a fast estimation error rate guarantee.

MASA's adaptive sampling substantially improves the quality of estimation for API shifts. In extensive experiments on real world ML APIs, MASA's assessment error is often an order of magnitude smaller than that of standard uniform sampling with same sample size (e.g., Figure 1 (c)). To reach the same tolerable target estimation error, MASA can reduce the required sample size by more than 50%, sometimes up to 90%.

**Contributions.** In short, our main contributions include:

1. We demonstrate that commercial Ml APIs can experience significant performance shifts over time, and formulate ML API shift assessments via confusion matrix difference estimation as an important practical problem. We will release the dataset consisting of 1,224,278 samples annotated by ML APIs in different date to stimulate more research on ML API shifts.

2. We propose MASA, an algorithm to assess the ML API performance shifts efficiently. MASA adaptively determines querying the ML API on which data points to minimize the shift estimation error under a sample size constraint. We show that MASA enjoys a low computation cost and performance guarantees.

3. We evaluate MASA on real world APIs from Google, Microsoft, Amazon and other providers for tasks including speech recognition, sentiment analysis, and facial emotion recognition. MASA leads to estimation errors an order of magnitude smaller than standard uniform sampling using the same sample size, or over 90% fewer samples to reach the same tolerable estimation error. Our code and datasets are also released [1].

**Related Work.  Distribution shifts in ML deployments:** Performance shifts in ML systems have been observed in applications like disease diagnosis (Lipton et al., 2018), facial recognition (Wang et al., 2020), and Inference of molecular structure (Koh et al., 2020). Most of them are attributed to distribution shifts, i.e., the distribution of the test and training datasets are different. Distribution shifts are usually modeled as covariate shifts (Shimodaira, 2000; Sugiyama et al., 2007; Quiñonero-Candela et al., 2009) or label shifts (Lipton et al., 2018; Saerens et al., 2002; Azizzadenesheli et al., 2019; Zhao et al., 2021). API shifts are orthogonal to distribution shifts: instead of attributing the performance shifts to data distribution changes, API shifts concern with ML APIs changes which changes its predictions on the same dataset. The methods for detecting distribution drifts typically rely on changes in data feature statistics and can not detect changes in the API on the same data. To the best of our knowledge, MASA is the first work to systematically investigate ML API shifts.

---

[1] https://github.com/lchen001/MASA

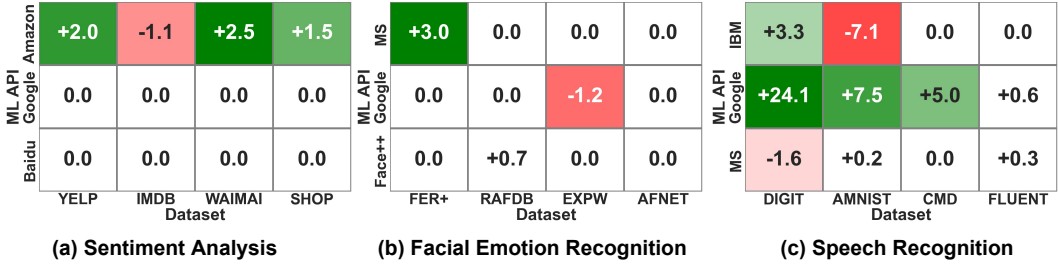

Figure 2: Observed overall accuracy changes. Each row corresponds to an ML API, and each column represents a dataset. The entry is the overall accuracy difference between evaluation in spring 2020 and spring 2021. In 12 out of 36 cases, the API's overall accuracy changed by more than 1%; this includes several cases of substantial drops in performance.

**Deploying and monitoring ML APIs:** Several issues in deployed ML APIs have been studied. For example, Buolamwini & Gebru (2018) showed that strong biases toward minority can exist in commercial APIs and Ribeiro et al. (2020) revealed that several bugs in commercial APIs can be detected using checklists. Kang et al. (2020) extend program assertions to monitor and improve deployed ML models. Chen et al. (2020) consider the trade-offs between accuracy performance and cost via exploiting multiple APIs. On the other hand, the proposed MASA focuses on estimating (silent) API performance changes cheaply and accurately, which has not been studied before.

**Stratified sampling and multi-arm bandits:** Stratified sampling has proved to be useful in various domains, such as approximate query processing (Chaudhuri et al., 2007), population mean estimation (Carpentier et al., 2011; 2015), and complex integration problems (Leprêtre et al., 2017). A common approach is to model stratified sampling as a multi-arm bandit (MAB) problem: view each data partition as an arm, and set the regret as the variance of the obtained estimator. Plenty of algorithms exist for solving standard MAB problems, such as the epsilon greedy algorithm (Slivkins, 2019), upper-confidence-bound approach (Auer et al., 2002), and Thompson sampling method (Russo et al., 2018). While those algorithms aim at playing the best arm as often as possible, stratified sampling's goal is often to estimate the average of all arms, and thus needs to allocate the number of calls for all arms aptly based on their variance. In contrast to estimating the population average in standard stratified sampling, our goal is to simultaneously estimate a matrix whose entries can be correlated. Thus, we have an objective (uncertainty score) that is different from theirs (variance), which also leads to an optimal allocation different from that of standard stratified sampling. This difference requires additional statistical bounds to prove convergence, which we provide in our paper. It also results in a different upper-confidence-bound term and convergence rate compared to standard bandit algorithms.

## 2 THE API SHIFT PROBLEM

**Empirical assessment of ML API shifts.** We start by making an interesting observation: *Commercial ML APIs' performance can change substantial over time on the same datasets*. We investigated twelve standard datasets across three different tasks, namely, YELP (Dat, c), IMDB (Maas et al.), WAIMAI (Dat, b), SHOP (Dat, a) for sentiment analysis, FER+ (Goodfellow et al., 2015), RAFDB (Li et al.), EXPW (Zhang et al.), AFNET (Mollahosseini et al., 2019) for facial emotion recognition, and DIGIT (Dat, d), AMNIST (Becker et al., 2018), CMD (Warden, 2018), FLUENT (Lugosch et al.), for speech recognition. For each dataset, we evaluated three commercial ML APIs' accuracy in April 2020 and April 2021. Figure 2 summarizes the overall accuracy changes.

There are several interesting empirical findings. First, API performance changes are quite common. In fact, as shown in Figure 2, API performance changes exceeding 1% occurred in about 33% of all (36) considered ML API-dataset combinations. Since the data distribution remains fixed, such a change is due to ML APIs' updates. Second, the API updates can either help or hurt the accuracy performance depending on the datasets. For example, as shown in Figure 2 (a), the Amazon sentiment analysis

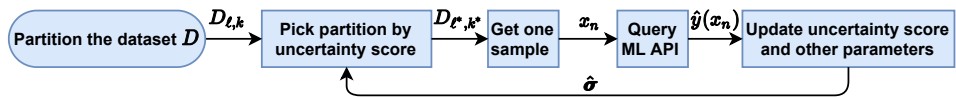

Figure 3: How MASA works. MASA first partitions the dataset. Then it picks which partition to sample based on some uncertainty measurement, queries the ML API on the drawn sample, and uses the API' prediction to update uncertainty and estimated shifts on this partition. This is repeated until the ML API has been queried $N$ times. Finally, the estimated shifts on different partitions are aptly fused to obtain the desired API shifts.

API's accuracy increases on YELP, WAIMAI, and SHOP, but decreases on IMDB. In addition, the update of Microsoft facial emotion recognition API only affects performance on the FER+ dataset, as shown in Figure 2 (b). Another interesting finding is that the magnitude of the performance change can be quite different. In fact, most of the accuracy differences are between 1–3%, but on DIGIT dataset, Google's accuracy change is more than 20%.

**Fine-grained assessment of API shift as changes in the confusion matrix.** Based on feedback from practitioners, accuracy change alone is insufficient, and attribution to per class change is often much more informative (Tsipras et al., 2020; Hou et al., 2019; Luque et al., 2019). Thus, a natural idea is to quantify an ML API's performance by its confusion matrix. We assess the change of the confusion matrix over time as a measure of API shift.

Formally, consider an ML service for a classification task with $L$ labels. For a data point $x$ from some domain $\mathcal{X}$, let $\hat{y}(x) \in [L]$ denote its predicted label on $x$, and $y(x)$ be the true label. For example, for sentiment analysis, $x$ is a text paragraph, and the task is to predict if the polarity of $x$ is positive or negative. Here $L = 2$, and $\hat{y}(x) = 1$ implies positive predicted label while $y(x) = 2$ indicates negative true label. The confusion matrix is denoted by $\boldsymbol{C} \in \mathbb{R}^{L \times L}$ where $\boldsymbol{C}_{i,j} \triangleq \Pr[y(x) = i, \hat{y}(x) = j]$. Given a confusion matrix of the ML API measured previously (say, a few months ago), $\boldsymbol{C}^o$, the ML API shift is defined as $\Delta \boldsymbol{C} \triangleq \boldsymbol{C} - \boldsymbol{C}^O$.

Using confusion matrix difference to quantify the ML API shift is informative. E.g, the overall accuracy change is simply the trace of $\Delta \boldsymbol{C}$. It also explains which label gets harder or easier for the updated API. Still consider, e.g., sentiment analysis. Given a 2% overall accuracy change, $\Delta \boldsymbol{C}_{1,2} = 1\%$ and $\Delta \boldsymbol{C}_{2,1} = -3\%$ implies that the change is due to predicting less (-3%) negative texts as positive, by sacrificing the accuracy on positive texts slightly (1%). This suggests that the API could have been updated with more negative training texts.

## 3 MASA: ML API Shift Assessment

Now we present MASA, an algorithmic framework efficiently to assess ML API shifts. Suppose the old confusion matrix $\boldsymbol{C}^o$ and a large labeled dataset $D$ are available. Given a query budget $N$, our goal is to generate $\Delta \hat{\boldsymbol{C}}$, an estimation of the API shifts as accurately as possible by querying the ML API $\hat{y}(\cdot)$ on $N$ samples drawn from $D$.

MASA achieves its goal via an adaptive sampling approach (Figure 3). It first divides the given dataset $D$ into several partitions (clusters). Then it adaptively decides which sample to query the ML API in an iterative manner: at each iteration, it selects one data partition based on some uncertainty measure (defined below), and queries the ML API on one sample randomly drawn from this partition. The API's prediction is obtained to update the uncertainty measure as well as the estimated shift $\Delta \hat{\boldsymbol{C}}$. This process is repeated until the ML API has been queried $N$ times or if a stopping rule is reached. We explain each step in detail next.

### 3.1 Data Partitioning

A key intuition in MASA is that not all samples are equally informative for estimating API shifts. Consider, for example, a vision API makes perfect predictions on "dog" images, and guesses randomly on "cat" pictures. The "dog" images are less informative, as even a small sample of "dog" queries

would tell that there is essentially no confusion for this class. Intuitively, within a sample budget, an estimator with more samples from "cat" pictures should be more accurate overall than that from "dog". Generally, harder images tend to be more informative.

Thus, it is a natural idea to partition all data points based on factors that may correlate with their informativeness, and sample from those partitions separately. In MASA, we use partitions $D_{i,k}$ that each contain the points with true label $i$ and difficulty level $k$. The difficulty level is an integer indicating how hard it is to predict the data point's label. It needs not be perfect, and can be simply the discretized prediction confidence generated by some simple ML models. A total of $L$ labels and $K$ distinct difficulty labels lead to a total of $LK$ partitions. If the uncertainty or variability of the ML API's prediction on each partition is different, then drawing a different number of samples from each partition may improve the shift assessment performance compared to standard uniform sampling. We verify this empirically in our evaluation (Section 4).

## 3.2 BUDGET ALLOCATION PROBLEM

Given the data partition, two questions arise: (i) how many samples should be drawn from each partition, and (ii) how to estimate the ML API shifts given available samples. The second question is relatively straightforward. Note that the API shifts satisfy

$$\Delta \boldsymbol{C}_{i,j} = \Pr[y(x) = i, \hat{y}(x) = j] - \boldsymbol{C}_{i,j}^o = \sum_{i=1}^{L} \sum_{k=1}^{K} \Pr[y(x) = i, \hat{y}(x) = j, x \in D_{i,k}] - \boldsymbol{C}_{i,j}^o$$

$$= \sum_{k=1}^{K} \Pr[\hat{y}(x) = j, x \in D_{i,k}] - \boldsymbol{C}_{i,j}^o = \sum_{k=1}^{K} \Pr[x \in D_{i,k}] \Pr[\hat{y}(x) = j | x \in D_{i,k}] - \boldsymbol{C}_{i,j}^o$$

where the first equation is by definition, the second is due to total probability rule, the third uses the fact that $x \in D_{i,k}$ implies $y(x) = i$, and the last equation applies conditional probability. Here, $\Pr[x \in D_{i,k}]$ is simply ratio of size of partition $D_{i,k}$ and entire dataset $D$, known a prior. To assess $\Delta \boldsymbol{C}_{i,j}$, we only need to estimate $\Pr[\hat{y}(x) = j | x \in D_{i,k}]$, the predicted label distribution on partition $D_{i,k}$. It can be estimated simply via the frequency of predicting label $j$ among all available samples drawn from $D_{i,k}$.

Now we consider the sample allocation problem. For ease of notation, we denote $\Pr[x_i \in D_{i,j}]$ by $\boldsymbol{p}_{i,k}$, $\Pr[\hat{y}(x) = j | x \in D_{i,k}]$ and its estimation by $\boldsymbol{\mu}_{i,k,j}$ and $\hat{\boldsymbol{\mu}}_{i,k,j}$, respectively. Then for deterministic sample allocations, the squared Frobenius norm error can be written as

$$\mathbb{E}\left[\|\Delta \boldsymbol{C} - \Delta \hat{\boldsymbol{C}}\|_F^2\right] = \sum_{i,j} \mathbb{E}\left(\Delta \boldsymbol{C}_{i,j} - \Delta \hat{\boldsymbol{C}}_{i,j}\right)^2 = \sum_{i,j} \mathbb{E}\left(\sum_k \boldsymbol{p}_{i,k}[\boldsymbol{\mu}_{i,k,j} - \hat{\boldsymbol{\mu}}_{i,k,j}]\right)^2$$

$$= \sum_{i,j,k} \boldsymbol{p}_{i,k}^2 \mathbb{E}\left([\boldsymbol{\mu}_{i,k,j} - \hat{\boldsymbol{\mu}}_{i,k,j}]\right)^2$$

Thus we use the loss $\mathcal{L}(\mathcal{A}, N) \triangleq \sum_{i,j,k} \boldsymbol{p}_{i,k}^2 \mathbb{E}\left([\boldsymbol{\mu}_{i,k,j} - \hat{\boldsymbol{\mu}}_{i,k,j}]\right)^2$ to measure the performance of any sample budget allocation algorithm $\mathcal{A}$ using $N$ samples. For any fixed $N$, our goal is to find a sample budget allocation algorithm $A$ to minimize the loss $\mathcal{L}(A, N)$. Notably, we can generalize it for other scenarios by replacing $(\Delta \boldsymbol{C} - \Delta \hat{\boldsymbol{C}})$ with $\boldsymbol{W} \odot (\Delta \boldsymbol{C} - \Delta \hat{\boldsymbol{C}})$, where $\odot$ is element-wise multiplication and $\boldsymbol{W}$ is an $L \times L$ weight matrix. Different choices of $\boldsymbol{W}$ can penalize the error of each entry in $\Delta \hat{\boldsymbol{C}}$ differently and serve for different purposes. For example, if misclassifying label 1 as label 2 is the only focus, then we can simply set $\boldsymbol{W}_{1,2} = 1$ and $\boldsymbol{W}_{i,j} = 0, \forall(i, j) \neq (1, 2)$. If we use the identity matrix as the weight $\boldsymbol{W}$, then it becomes equivalent to estimating the overall accuracy by minimizing the variance. To minimize the loss for the general scenarios, Algorithm 1 (which is explained in the rest of this section) is still applicable by simply multiplying the API selection formula (line 3 in Algorithm 1) with its corresponding weights.

## 3.3 UNCERTAINTY SCORE AND OPTIMAL ALLOCATION

The optimal sample allocation is directly connected to how informative each data partition is. To see this, let us first introduce the notation of *uncertainty score* for each data partition.

---

**Algorithm 1** MASA's ML API shift assessment algorithm.

---

**Input** : ML API $\hat{y}(\cdot)$, query budget $N$, partitions $D_{i,k}, \boldsymbol{p} \in \mathbb{R}^{L \times K}, \boldsymbol{C}^o \in \mathbb{R}^{L \times L}$, and $a > 0$

**Output** : Estimated ML API Shift $\Delta \hat{\boldsymbol{C}} \in \mathbb{R}^{L \times \hat{L}}$

1 Set $\boldsymbol{N} = \boldsymbol{0}_{L \times K}, \hat{\boldsymbol{\mu}} = \boldsymbol{0}_{L \times K \times L}, \hat{\boldsymbol{\sigma}} = \boldsymbol{0}_{L \times K}, \boldsymbol{H} = \boldsymbol{0}_{L \times K \times L}$       ▷ Initialization

2 **for** $n \leftarrow 1$ **to** $N$ **do**

3    $(i^*, k^*) \leftarrow \begin{cases} (i, k), & \text{if } \boldsymbol{N}_{i,k} < 2 \\ \arg\max_{i,k} \frac{\boldsymbol{p}_{i,k}}{\boldsymbol{N}_{i,k}} \left( \hat{\boldsymbol{\sigma}}_{i,k} + \sqrt[4]{\frac{a}{\boldsymbol{N}_{i,k}}} \right), & o/w \end{cases}$     ▷ Determine data partition

4    Sample $x_n$ from $D_{i^*,k^*}$ and query the ML API to obtain $\hat{y}(x_n)$

5    $\boldsymbol{N}_{i^*,k^*} \leftarrow \boldsymbol{N}_{i^*,k^*} + 1$       ▷ Update sample size

6    $\hat{\boldsymbol{\mu}}_{i^*,k^*,j} \leftarrow \hat{\boldsymbol{\mu}}_{i^*,k^*,j} + \frac{\mathbb{1}_{\hat{y}(x_n)=j} - \hat{\boldsymbol{\mu}}_{i^*,k^*,j}}{\boldsymbol{N}_{i^*,k^*}}, \forall j \in [L]$     ▷ Update predicted label distribution

7    $\hat{\boldsymbol{\sigma}}^2_{i^*,k^*} \leftarrow \begin{cases} \frac{1}{2} \boldsymbol{H}_{i^*,k^*,\hat{y}(x_n)}, & \text{if } \boldsymbol{N}_{i^*,k^*} < 2 \\ \hat{\boldsymbol{\sigma}}^2_{i^*,k^*} + \frac{1 - \frac{\boldsymbol{H}_{i^*,k^*,\hat{y}(x_n)}}{\boldsymbol{N}_{i^*,k^*}-1} - \hat{\boldsymbol{\sigma}}^2_{i^*,k^*}}{\boldsymbol{N}_{i^*,k^*}}, & o/w \end{cases}$     ▷ Update uncertainty score

8    $\boldsymbol{H}_{i^*,k^*,\hat{y}(x_n)} \leftarrow \boldsymbol{H}_{i^*,k^*,\hat{y}(x_n)} + 1$       ▷ Update label frequency

9 **end**

10 Return $\Delta \hat{\boldsymbol{C}} \in \mathbb{R}^{L \times L}$ where $\Delta \hat{\boldsymbol{C}}_{i,j} = \sum_{k=1}^{K} \boldsymbol{p}_{i,k} \hat{\boldsymbol{\mu}}_{i,k,j} - \boldsymbol{C}^o_{i,j}, \forall i, j$     ▷ Confusion estimation

---

**Definition 1.** $\boldsymbol{\sigma}^2_{i,k} \triangleq (1 - \sum_{j=1}^{L} \Pr^2[\hat{y}(x) = j | x \in D_{i,k}])$ *denotes the uncertainty score of* $D_{i,k}$.

The uncertainty score quantifies how informative each $D_{i,k}$ is by subtracting from 1 the sum of the square of each label's probability mass. The uncertainty score is related to collision entropy (discussed in Appendix A), and determines the optimal allocation as follows.

**Lemma 1.** *Let* $A^*$ *be the sample allocation algorithm that achieves the smallest expected squared Frobenius norm error. Then the number of samples drawn from* $D_{i,k}$ *by* $A^*$ *is*

$$\boldsymbol{N}^*_{i,k} = \frac{\boldsymbol{p}_{i,k} \boldsymbol{\sigma}_{i,k}}{\sum_{i,k} \boldsymbol{p}_{i,k} \boldsymbol{\sigma}_{i,k}} N$$

Lemma 1 shows that the optimal budget allocation depends on the uncertainty score, but in practice, we do not know the uncertainty score before drawing samples and querying the ML API. Thus, a natural question is how to estimate the uncertainty score $\boldsymbol{\sigma}^2_{i,k}$. Suppose $n$ samples, $x_1, x_2, \cdots, x_n$, are drawn from partition $D_{i,k}$. Then we can estimate $\boldsymbol{\sigma}^2_{i,k}$ by

$$\hat{\boldsymbol{\sigma}}^2_{i,k} \triangleq 1 - \frac{1}{n(n-1)} \sum_{s=1}^{n} \sum_{t:t=1, t \neq s}^{n} \mathbb{1}_{\hat{y}(x_s) = \hat{y}(x_t)} \tag{3.1}$$

### 3.4 An Uncertainty-aware Adaptive Sampling Algorithm

Now we have a chicken-and-egg problem: estimating the uncertainty scores is needed to find the optimal sample allocation, but sampling from all partitions is needed to estimate their uncertainty scores. To overcome this issue, we adopt an iterative sampling approach, as shown in Algorithm 1. At each iteration, it alternates between (i) uncertainty score-based new sample selection (line 3 - 4) and (ii) uncertainty score and predicted label distribution update using the new sample (line 5 - 8). After querying the ML API $N$ times, the API shifts are obtained by (iii) fusing the estimated predicted label distribution on each partition (line 10). We give the details as follows.

**Uncertainty score and predicted label distribution update.** After obtaining the predicted label for a sample from partition $D_{i^*,k^*}$, we need to update (i) the number of samples already drawn from this partition, denoted by $\boldsymbol{N}_{i^*,k^*}$, (ii) the estimated predicted label distribution, denoted by $\hat{\boldsymbol{\mu}}_{i^*,k^*,j}, \forall j$, and (iii) the estimated uncertainty score, $\hat{\boldsymbol{\sigma}}^2_{i^*,k^*}$. For $\boldsymbol{N}_{i^*,k^*}$ and $\hat{\boldsymbol{\mu}}_{i^*,k^*,j}$ (line 5-6), we use standard incremental update approach (Cotton, 1975), which requires constant space and computational cost per iteration. For $\hat{\boldsymbol{\sigma}}^2_{i^*,k^*}$, we additionally maintain the number of label $j$ being predicted among all samples drawn from $D_{i^*,k^*}$, denoted by $\boldsymbol{H}_{i^*,k^*,j}$ (line 8). This enables a fast incremental update of $\hat{\boldsymbol{\sigma}}^2_{i^*,k^*}$ (line 7).

**Uncertainty score-based new sample selection.** To determine on which partition to select a new sample, we use an upper-confidence-bound approach on the weighted uncertainty score (second case in line 3), after ensuring two samples have been drawn from each partition (first case in line 3). Two samples are needed for an initial estimation of each partition's uncertainty score. Here, we use a parameter $a > 0$ to balance between exploiting knowledge of uncertainty score ($\hat{\sigma}_{i,k}^2$) and exploring more partitions ($\sqrt[4]{\frac{1}{N_{i,k}}}$).

We quantify the performance of MASA v.s. the optimal allocation algorithm $A^*$ as follows.

**Theorem 2.** *If $a > 2 \log L + \log K + \frac{9}{4} \log N$ and $N > 4LK$, then we have*

$$\mathcal{L}(\text{MASA}, N) - \mathcal{L}(\mathcal{A}^*, N) \leq O(N^{-\frac{5}{4}} \log^{\frac{1}{4}} N)$$

Roughly speaking, Theorem 2 shows that the loss gap between the API shift estimated by MASA and the (unreachable) optimal allocation algorithm ceases in the rate of $N^{-5/4}$. Note that the loss of the optimal allocation decays in the rate of $N^{-1}$. Thus, as $N$ gets larger and larger, the relative gap becomes more and more negligible.

## 4  EXPERIMENTS

We apply MASA to estimate the shifts of several real world ML services for various tasks. Our goal is three-fold: (i) understand if and why MASA assess the API shifts efficiently, (ii) examine how much sample cost MASA can reduce compared to standard sampling, and (iii) exploit the trade-offs between estimation accuracy and query cost achieved by MASA. We also study how the hyperparameters affect MASA's performance, left to Appendix C.

**Tasks, ML APIs, and datasets.** As shown in Section 2, we have observed 12 of 36 cases where there is a >1% overall accuracy performance change of an ML API. Thus, we focus on evaluating MASA's performance on those 12 cases. Except for case study, all experiments were averaged over 1500 runs. In all tasks, we created partitions using difficulty levels induced by a cheap open source model from GitHub. More details are in Appendix C.

**Sentiment analysis: a case study on Amazon API.** We start by a case study on Amazon API on a sentiment analysis dataset, YELP to understand MASA's performance. We adopt MASA with sample budget 2000. The dataset is divided into 4 partitions $D_{+,l}, D_{+,h}, D_{-,l}, D_{-,h}$, depending on whether the true label is positive (+) or negative (-), and quality score produced by the 2020 version is lower (l) or higher (h) than the median.

We first note that the API shift gives an interesting explanation to Amazon API's accuracy change. In fact, as shown in Figure 4 (a-c), the accuracy increase is mostly because more texts (2.7%) with negative altitudes are correctly classified. One possible explanation is that the API has been retrained on a dataset with more negative texts. Next, we observe that MASA produces accurate estimation of the API shift by comparing Figure 4 (c) and (d). This is primarily due to (i) that the data partitioning separates more uncertain data from less uncertain ones, and (ii) that adaptive sampling learns the uncertainty level effectively. Figure 4 (e-f) shows that while the partitions' size is similar, their uncertainty scores are different. For example, higher quality score implies a much smaller uncertainty for positive texts ($D_{+,h}$ and $D_{+,l}$ in Figure 4(f)). As shown in Figure 4 (g), MASA indeed learns to utilize such imbalanced uncertainty: its sampling allocation for each partition is quite close to the optimal allocation (dark star point). Finally, it is worth noting that MASA outperforms standard uniform sampling notably, as shown in Figure 4(h). This is because uniform sampling does not exploit the uncertainty of each partition.

**Budget savings achieved by MASA.** In many applications, it suffices to obtain an estimated API shift close to the true shift, e.g., within a 1% Frobenius norm error. Thus, a natural question arises: *to reach the same tolerable estimation error, how much sampling cost can* MASA *reduce compared to standard sampling approaches?*

To answer this question, we compare MASA with two natural approaches: (i) uniform sampling and (ii) stratified sampling by drawing same number of samples for each true label. For each approach,

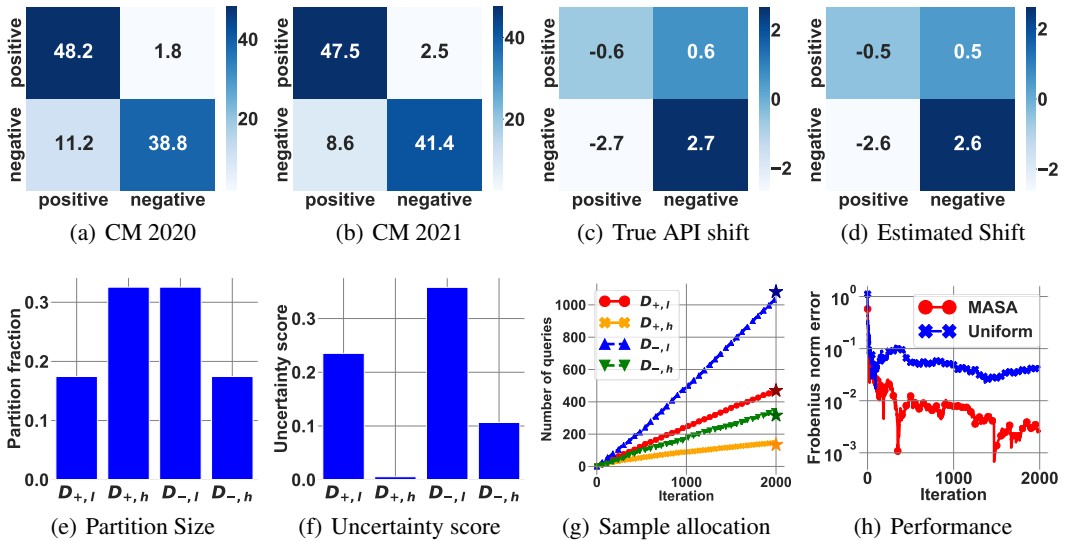

Figure 4: Case study for Amazon API's performance shift on dataset YELP. (a) and (b) give its confusion matrix in spring 2020 and spring 2021, respectively. (c) is their differences, i.e., the API shift. MASA's estimated shift using 2000 samples is in (d). The dataset is divided into 4 partitions based on (i) positive ($+$) or negative ($-$) true labels, and (ii) low ($l$) or high ($h$) quality score. (e) and (f) give the size and uncertainty score of each partitions. (g) shows MASA's sampling decision per iteration, where the dark dot points represent the (unreachable) optimal sample allocation. (h) reveals its performance.

Table 1: Required sample size to reach 1% Frobnius norm error. Here we compare MASA with uniform (U) sampling and stratified (S) sampling. U and S required very similar sample sizes and are reported in the same column. The sample size is obtained when a 1% Frobenius norm error is achieved with probability 95%.

| API;Dataset | Sample size | | Save | API;Dataset | Sample size | | Save |
|---|---|---|---|---|---|---|---|
| | MASA | U/S | | | MASA | U/S | |
| Amazon;YELP | 4.5K | 19.7K | 77% | IBM;DIGIT | 3.6K | 17.0K | 79% |
| Amazon;IMDB | 10.3K | 20.8K | 51% | IBM;AMNIST | 2.4K | 18.5K | 87% |
| Amazon;WAIMAI | 7.8K | 18.0K | 57% | Google;DIGIT | 4.2K | 17.0K | 75% |
| Amazon;SHOP | 4.8K | 20.8K | 77% | Google;AMNIST | 1.1K | 18.5K | 94% |
| MS; FER+ | 2.6K | 19.9K | 87% | Google;CMD | 1.6K | 15.2K | 89% |
| Google; EXPW | 4.2K | 17.9K | 77% | MS;DIGIT | 3.3K | 17.0K | 81% |

we measure the number of samples needed to reach 1% Frobenius norm error with probability 95%, via an upper bound on the estimated Frobenius error. The details are left to Appendix C. As shown in Table 1, MASA usually requires more than 70% fewer samples to reach such tolerable Frobenius norm error than the uniform and stratified sampling, primarily due to its shift estimation is more accurate. Uniform and stratified sampling required the similar number of samples because the upper bounds on their estimated Frobenius error are similar.

**Trade-offs between estimation error and query budget .** Next we examine the trade-offs between API shift estimation error and sample size achieved by MASA, shown in Figure 5. We first note that, across all 12 observed API shifts, MASA consistently outperforms standard uniform sampling for any fixed sample size. In fact, the achieved estimation error of MASA is usually an order of magnitude smaller than that of uniform sampling. This verifies that MASA can provide more accurate

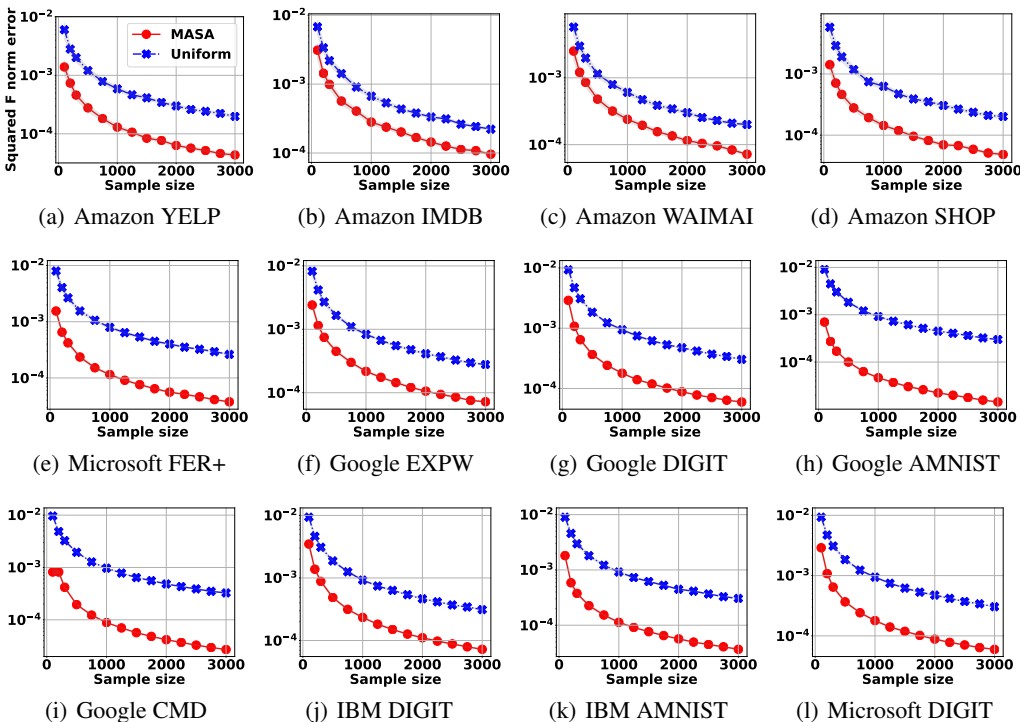

Figure 5: API shift estimation performance and sample size trade-offs. We compare the expected squared Frobenius norm error of MASA with $K = 3$ partitions versus standard uniform sampling. For any sample size, MASA consistently leads to an estimation error much smaller than uniform sampling across different API and dataset combinations.

assessments of API shifts in diverse applications. Second, some API shifts are easier to estimate than others. For example, for Google API shift on AMNIST, using 1000 samples already gives an expected squared Frobenius norm error lower than $10^{-4}$, while it usually requires 2000 samples for other shifts. This is probably because the skew in its uncertainties among different partitions is more severe than other shifts.

## 5 CONCLUSION

In this paper, we identify and formulate the problem of characterizing ML API shifts. Our systematic empirical study has shown that API model updates are frequent, and that some updates can reduce performance substantially. Quantifying such shifts is an important but understudied problem that can greatly affect the reliability of applications using ML-as-a-service. To capture fine-grained change assessment, we model the ML API shifts as confusion matrix differences. Adaptive sampling methods are typically designed for estimating a single scalar (e.g., adaptive population mean) or training a model (e.g., active learning), and thus not directly applicable for API shift estimation. Uniform sampling is natural but requires a large number of samples. Thus, we propose an algorithmic framework, MASA, to adaptively assess the API shifts using as few samples as possible. Our work focuses on estimating changes in the confusion matrix because the confusion matrix is often what is used by practitioners to assess API performance. We acknowledge that confusion matrices are most applicable for classification tasks, and other measures need to be used for more complex APIs (e.g. OCR, NLP). While this is a limitation, classification with a small to moderate number of classes of interest is a common use case for ML APIs, and this is an important starting point since it has not been studied before. We also release our dataset of 1,224,278 samples annotated by commercial APIs in different dates as the first dataset and resource for studying ML API performance shifts.

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

## ACKNOWLEDGEMENT

This work was supported in part by a Google PhD Fellowship, NSF CCF 1763191, NSF CAREER 1651570 and 1942926, NIH P30AG059307, NIH U01MH098953, grants from the Chan-Zuckerberg Initiative, Sutherland, and affiliate members and other supporters of the Stanford DAWN project—Ant Financial, Meta, Google, Infosys, NEC, and VMware—as well as Cisco and SAP. We also thank anonymous reviewers for helpful discussion and feedback.

**Appendix Outline**   The appendix is organized as follows. Section A provides additional technical details. All proofs are presented in Section B. We give experimental setups, details of datasets and ML APIs, and further empirical results in Section C.

## A   TECHNICAL DETAILS

**Computation and space cost of MASA.**   One attractive property of MASA is its low computation and space cost. In fact, it can be easily verified from Algorithm 1 that, the computation cost is only linear in the number of samples $N$. The occupied space is only constant. Therefore, MASA can be easily applied for large number of samples.

**Choice of parameter $a$.**   The parameter $a$ is used to balance between exploiting and exploration in Algorithm 1. Throughout this paper, we set $a = 1$ as the default value. While theoretically $a$ should depend on the partition size and sample number, in practice we found that $a = 1$ works well. An in-depth analysis for this remains an interesting open problem.

**Stopping rule under loss requirements.**   For MASA, we establish the upper bound on the loss by (i) computing the upper bound on the estimated uncertainty score for each partition, and (ii) summing up all those upper bounds weighted by the partition size to form the upper bound on the loss. For Uniform sampling or stratified sampling, we directly use the upper bound on the Frobenius loss. Here, we adopt the standard upper bound for Bernoulli variables. That is to say, for any estimator using $n$ samples, we use $\sqrt{\frac{c}{n}}$ as its upper bound, where $c$ is a parameter to control the confidence. For both methods, we choose $c$ to ensure a 1% error under 95% confidence level.

## B   PROOFS

We present all missing proofs here. For ease of expositions, let us first introduce a few notations. We let $x_n$ denote the $n$th sample drawn in Algorithm 1, and use $I_n$ to indicate from which partition the sample $x_n$ is drawn. For example, $I_n = (i, k)$ indicates that $x_n$ is drawn from the partition $D_{i,k}$.

Let $\boldsymbol{z}_{\ell,k,t} \in [L]$ denote the ML API's predicted label for the $t$th sample drawn from the partition $D_{\ell,k}$. Abusing the notation a little bit, let $\boldsymbol{N}_{\ell,k,n}$ denote the value of $\boldsymbol{N}_{\ell,k}$ after the $n-1$th iteration and before the $n$th iteration in Algorithm 1. Similarly, let $\hat{\boldsymbol{\sigma}}_{\ell,k,n}$ be the value of $\hat{\boldsymbol{\sigma}}_{\ell,k}$, $\hat{\boldsymbol{\mu}}_{\ell,k,j,n}$ be the value of $\hat{\boldsymbol{\mu}}_{\ell,k,j}$, and $\boldsymbol{H}_{\ell,k,,j,n}$ be the value of $\boldsymbol{H}_{\ell,k,j}$, all after the $n-1$th iteration and before the $n$th iteration in Algorithm 1. In addition, let $\Delta_{\ell,k} \triangleq \frac{\boldsymbol{p}_{\ell,k}\boldsymbol{\sigma}_{\ell,k}}{\sum_{\ell',k'} \boldsymbol{p}_{\ell',k'}\boldsymbol{\sigma}_{\ell',k'}}$ and $\Delta_{\min} = \min \Delta_{\ell,k}$. Similarly, let us denote $\boldsymbol{\sigma}_{\min} \triangleq \min \boldsymbol{\sigma}_{\ell,k}$. By assumption that $\boldsymbol{p}_{\ell,k} > 0$ and $\boldsymbol{\sigma}_{\ell,k} > 0$, we must have $\Delta_{\min} > 0$ and $\boldsymbol{\sigma}_{\min} > 0$.

### B.1   USEFUL LEMMAS

Let us first give a few useful lemmas. The first gives a high probability bound on our estimated uncertainty score.

**Lemma 3.** *Let the event $A$ be*

$$A \triangleq \bigcap_{\substack{1 \le k \le K, 1 \le \ell \le L \\ 1 \le t \le N}} \left\{ \left| \sqrt{1 - \frac{1}{t(t-1)} \sum_{i=1}^{t} \sum_{j=1, j \neq i}^{t} \mathbb{1}_{\boldsymbol{z}_{\ell,k,i} = \boldsymbol{z}_{\ell,k,j}}} - \boldsymbol{\sigma}_{\ell,k} \right| \le \sqrt[4]{\frac{\log 2/\delta}{2t}} \right\}$$

*Then for any $\delta > 0$, we have $\Pr[A] \ge 1 - LKN\delta$.*

*Proof.* For any fixed $t$, let us first denote

$$f(\boldsymbol{z}_{\ell,k,1}, \boldsymbol{z}_{\ell,k,2}, \cdots, \boldsymbol{z}_{\ell,k,t}) \triangleq 1 - \frac{1}{t(t-1)} \sum_{i=1}^{t} \sum_{j=1,j\neq i}^{t} \mathbb{1}_{\boldsymbol{z}_{\ell,k,i}=\boldsymbol{z}_{\ell,k,j}}$$

Its expectation is simply

$$\begin{aligned}
\mathbb{E}[f(\boldsymbol{z}_{\ell,k,1}, \boldsymbol{z}_{\ell,k,2}, \cdots, \boldsymbol{z}_{\ell,k,t})] &= \mathbb{E}[1 - \frac{1}{t(t-1)} \sum_{i=1}^{t} \sum_{j=1,j\neq i}^{t} \mathbb{1}_{\boldsymbol{z}_{\ell,k,i}=\boldsymbol{z}_{\ell,k,j}}] \\
&= 1 - \frac{1}{t(t-1)} \mathbb{E}[\sum_{i=1}^{t} \sum_{j=1,j\neq i}^{t} \mathbb{1}_{\boldsymbol{z}_{\ell,k,i}=\boldsymbol{z}_{\ell,k,j}}] \\
&= 1 - \mathbb{E}[\mathbb{1}_{\boldsymbol{z}_{\ell,k,i}=\boldsymbol{z}_{\ell,k,j}}]
\end{aligned}$$

where the second equation applies the linearity of expectation, and the third equation uses the fact that all $\boldsymbol{z}_{\ell,k,i}$ are identically independent. Note that

$$\begin{aligned}
&\mathbb{E}[\mathbb{1}_{\boldsymbol{z}_{\ell,k,i}=\boldsymbol{z}_{\ell,k,j}}] \\
&= \Pr[\boldsymbol{z}_{\ell,k,i} = \boldsymbol{z}_{\ell,k,j}] \\
&= \sum_{r=1}^{L} \Pr[\boldsymbol{z}_{\ell,k,i} = r] \Pr[\boldsymbol{z}_{\ell,k,j} = r] \\
&= \sum_{r=1}^{L} \overset{2}{\Pr}[\boldsymbol{z}_{\ell,k,i} = r]
\end{aligned}$$

where the first equation uses the definition of indicator function, the second uses the fact that two sample are independent and there are only $L$ many possible labels, and the last equation uses the fact that those samples' distribution is identical. Applying this in the above equation, we get

$$\begin{aligned}
\mathbb{E}[f(\boldsymbol{z}_{\ell,k,1}, \boldsymbol{z}_{\ell,k,2}, \cdots, \boldsymbol{z}_{\ell,k,t})] &= 1 - \mathbb{E}[\mathbb{1}_{\boldsymbol{z}_{\ell,k,i}=\boldsymbol{z}_{\ell,k,j}}] \\
&= 1 - \sum_{r=1}^{L} \overset{2}{\Pr}[\boldsymbol{z}_{\ell,k,i} = r] = \boldsymbol{\sigma}_{\ell,k}^2
\end{aligned}$$

That is to say, its expectation is simply the uncertainty score $\boldsymbol{\sigma}_{\ell,k}^2$. On the other hand, we note that, for any $i$, we have

$$f(\boldsymbol{z}_{\ell,k,1}, \cdots, \boldsymbol{z}_{\ell,k,i-1}, \boldsymbol{z}_{\ell,k,i}, \boldsymbol{z}_{\ell,k,i+1}, \cdots, \boldsymbol{z}_{\ell,k,t}) - f(\boldsymbol{z}_{\ell,k,1}, \cdots, \boldsymbol{z}_{\ell,k,i-1}, \boldsymbol{z}'_{\ell,k,i}, \boldsymbol{z}_{\ell,k,i+1}, \cdots, \boldsymbol{z}_{\ell,k,t})$$

$$= \frac{1}{t(t-1)} \sum_{j=1,j\neq i}^{t} \mathbb{1}_{\boldsymbol{z}'_{\ell,k,i}=\boldsymbol{z}_{\ell,k,j}} - \mathbb{1}_{\boldsymbol{z}_{\ell,k,i}=\boldsymbol{z}_{\ell,k,j}} \leq \frac{1}{t(t-1)} \cdot (t-1) = \frac{1}{t}$$

where the inequality is due to the fact that the indicator function can only take values in $\{0,1\}$. Similarly, we have

$$f(\boldsymbol{z}_{\ell,k,1}, \cdots, \boldsymbol{z}_{\ell,k,i-1}, \boldsymbol{z}_{\ell,k,i}, \boldsymbol{z}_{\ell,k,i+1}, \cdots, \boldsymbol{z}_{\ell,k,t}) - f(\boldsymbol{z}_{\ell,k,1}, \cdots, \boldsymbol{z}_{\ell,k,i-1}, \boldsymbol{z}'_{\ell,k,i}, \boldsymbol{z}_{\ell,k,i+1}, \cdots, \boldsymbol{z}_{\ell,k,t})$$

$$= \frac{1}{t(t-1)} \sum_{j=1,j\neq i}^{t} \mathbb{1}_{\boldsymbol{z}'_{\ell,k,i}=\boldsymbol{z}_{\ell,k,j}} - \mathbb{1}_{\boldsymbol{z}_{\ell,k,i}=\boldsymbol{z}_{\ell,k,j}} \geq \frac{1}{t(t-1)} \cdot -(t-1) = -\frac{1}{t}$$

By Mcdiarmid inequality, we have

$$\Pr[|f(\boldsymbol{z}_{\ell,k,1}, \boldsymbol{z}_{\ell,k,2}, \cdots, \boldsymbol{z}_{\ell,k,t}) - \mathbb{E}[f(\boldsymbol{z}_{\ell,k,1}, \boldsymbol{z}_{\ell,k,2}, \cdots, \boldsymbol{z}_{\ell,k,t})]| \geq \epsilon] \leq 2e^{-\frac{2\epsilon^2}{\sum_{i=1}^{t} t^{-2}}} = 2e^{-2t\epsilon^2}$$

Set $\delta = 2e^{-2t\epsilon^2}$. This simply becomes, with probability at most $\delta$,

$$|f(\boldsymbol{z}_{\ell,k,1}, \boldsymbol{z}_{\ell,k,2}, \cdots, \boldsymbol{z}_{\ell,k,t}) - \boldsymbol{\sigma}_{\ell,k}^2|$$

$$= |f(\boldsymbol{z}_{\ell,k,1}, \boldsymbol{z}_{\ell,k,2}, \cdots, \boldsymbol{z}_{\ell,k,t}) - \mathbb{E}[f(\boldsymbol{z}_{\ell,k,1}, \boldsymbol{z}_{\ell,k,2}, \cdots, \boldsymbol{z}_{\ell,k,t})]| \geq \sqrt{\frac{\log 2/\delta}{2t}}$$

Note that $f$ is positive, we can take square root of both side, and obtain with probability at most $\delta$,

$$|\sqrt{f(\boldsymbol{z}_{\ell,k,1}, \boldsymbol{z}_{\ell,k,2}, \cdots, \boldsymbol{z}_{\ell,k,t})} - \boldsymbol{\sigma}_{\ell,k}| \geq \sqrt[4]{\frac{\log 2/\delta}{2t}}$$

Or alternatively, with probability at least $1 - \delta$,

$$|\sqrt{f(\boldsymbol{z}_{\ell,k,1}, \boldsymbol{z}_{\ell,k,2}, \cdots, \boldsymbol{z}_{\ell,k,t})} - \boldsymbol{\sigma}_{\ell,k}| \leq \sqrt[4]{\frac{\log 2/\delta}{2t}}$$

which holds for fixed $t, \ell, k$. Taking union bound, we know that with probability $1 - KLN\delta$,

$$|\sqrt{f(\boldsymbol{z}_{\ell,k,1}, \boldsymbol{z}_{\ell,k,2}, \cdots, \boldsymbol{z}_{\ell,k,t})} - \boldsymbol{\sigma}_{\ell,k}| \leq \sqrt[4]{\frac{\log 2/\delta}{2t}}$$

which holds for all $t, \ell, k$. Plugging in the form of $f$ completes the proof. $\qquad\square$

The next one is more technical: it gives a connection between stopping time and adaptive sampling. We omit the proof and refer the interested readers to (Athreya & Lahiri, 2006).

**Lemma 4** (Wald's second inequality). *Let $\{\mathcal{F}_t\}_{t=1,\ldots,n}$ be a filtration and $\{X_t\}_{t=1,\ldots,n}$ be an $\mathcal{F}_t$ adapted sequence of i.i.d. random variables with finite expectation $\mu$ and variance $Var$. Assume that $\mathcal{F}_t$ and $\sigma(\{X_s : s \geq t+1\})$ are independent for any $t \leq n$, and let $T(\leq n)$ be a stopping time with respect to $\mathcal{F}_t$. Then*

$$\mathbb{E}\left[\left(\sum_{i=1}^{T} X_i - T\,\mu\right)^2\right] = \mathbb{E}[T]\ Var\,.$$

## B.2 PROOF OF LEMMA 1

*Proof.* Recall that the loss, defined as the expected squared Frobenius norm error, is

$$\mathbb{E}\left[\|\Delta\boldsymbol{C} - \Delta\hat{\boldsymbol{C}}\|_F^2\right] = \sum_{i,j} \mathbb{E}\left(\Delta\boldsymbol{C}_{i,j} - \Delta\hat{\boldsymbol{C}}_{i,j}\right)^2 = \sum_{i,j} \mathbb{E}\left(\sum_k \boldsymbol{p}_{i,k}[\boldsymbol{\mu}_{i,k,j} - \hat{\boldsymbol{\mu}}_{i,k,j}]\right)^2$$

$$= \sum_{i,j,k} \boldsymbol{p}_{i,k}^2 \mathbb{E}\left([\boldsymbol{\mu}_{i,k,j} - \hat{\boldsymbol{\mu}}_{i,k,j}]\right)^2$$

Here we basically apply the definition of each entry. Suppose $\boldsymbol{N}_{i,k}$ samples are allocated to estimate $\boldsymbol{\mu}_{i,k,j}$. Then we have

$$\mathbb{E}\left([\boldsymbol{\mu}_{i,k,j} - \hat{\boldsymbol{\mu}}_{i,k,j}]\right)^2 = \frac{1}{\boldsymbol{N}_{i,k}} \Pr[\hat{y}(x) = j | x \in D_{i,k}](1 - \Pr[\hat{y}(x) = j | x \in D_{i,k}])$$

since $\boldsymbol{\mu}_{i,k,j}$ is effectively a Bernoulli variable. Then the loss becomes

$$\mathbb{E}\left[\|\Delta\boldsymbol{C} - \Delta\hat{\boldsymbol{C}}\|_F^2\right] = \sum_{i,j,k} \boldsymbol{p}_{i,k}^2 \mathbb{E}\left([\boldsymbol{\mu}_{i,k,j} - \hat{\boldsymbol{\mu}}_{i,k,j}]\right)^2$$

$$= \sum_{i,j,k} \boldsymbol{p}_{i,k}^2 \frac{1}{\boldsymbol{N}_{i,k}} \Pr[\hat{y}(x) = j | x \in D_{i,k}](1 - \Pr[\hat{y}(x) = j | x \in D_{i,k}])$$

$$= \sum_{i,k} \boldsymbol{p}_{i,k}^2 \frac{1}{\boldsymbol{N}_{i,k}} \sum_j \Pr[\hat{y}(x) = j | x \in D_{i,k}](1 - \Pr[\hat{y}(x) = j | x \in D_{i,k}])$$

where the last equation is simply by rearranging the summation. Note that

$$\sum_j \Pr[\hat{y}(x) = j | x \in D_{i,k}] = 1$$

The last summation is simply

$$\sum_j \Pr[\hat{y}(x) = j | x \in D_{i,k}](1 - \Pr[\hat{y}(x) = j | x \in D_{i,k}]) = 1 - \sum_j \overset{2}{\Pr}[\hat{y}(x) = j | x \in D_{i,k}])$$

$$= \boldsymbol{\sigma}_{i,k}^2$$

Thus, the loss becomes

$$\mathbb{E}\left[\|\Delta \boldsymbol{C} - \Delta\hat{\boldsymbol{C}}\|_F^2\right] = \sum_{i,k} \boldsymbol{p}_{i,k}^2 \boldsymbol{\sigma}_{i,k}^2 \frac{1}{\boldsymbol{N}_{i,k}}$$

By Cauchy Schwarz inequality, we have

$$\left(\sum_{i,k} \frac{\boldsymbol{p}_{i,k}^2 \boldsymbol{\sigma}_{i,k}^2}{\boldsymbol{N}_{i,k}}\right) \left(\sum_{i,k} \boldsymbol{N}_{i,k}\right) \geq \left(\sum_{i,k} \boldsymbol{p}_{i,k}\boldsymbol{\sigma}_{i,k}\right)^2$$

where the equality holds if and only if

$$\frac{\boldsymbol{p}_{i,k}^2 \boldsymbol{\sigma}_{i,k}^2}{\boldsymbol{N}_{i,k}^2} = \frac{\boldsymbol{p}_{i',k'}^2 \boldsymbol{\sigma}_{i',k'}^2}{\boldsymbol{N}_{i',k'}^2}$$

for any $i, i', k, k'$. That is to say, there exists some constant $c$, such that

$$\frac{\boldsymbol{p}_{i,k}\boldsymbol{\sigma}_{i,k}}{\boldsymbol{N}_{i,k}} = \frac{\boldsymbol{p}_{i',k'}\boldsymbol{\sigma}_{i',k'}}{\boldsymbol{N}_{i',k'}} = \frac{1}{c}$$

And thus, $\boldsymbol{N}_{i,k} = \boldsymbol{p}_{i,k}\boldsymbol{\sigma}_{i,k}c$. Summing over $i, k$ gives

$$N = \sum_{i,k} \boldsymbol{N}_{i,k} = \sum_{i,k} \boldsymbol{p}_{i,k}\boldsymbol{\sigma}_{i,k}c$$

Thus,

$$c = \frac{N}{\sum_{i,k} \boldsymbol{p}_{i,k}\boldsymbol{\sigma}_{i,k}}$$

and

$$\boldsymbol{N}_{i,k} = \boldsymbol{p}_{i,k}\boldsymbol{\sigma}_{i,k}c = \boldsymbol{p}_{i,k}\boldsymbol{\sigma}_{i,k} \cdot \frac{1}{\sum_{i,k} \boldsymbol{p}_{i,k}\boldsymbol{\sigma}_{i,k}} = \frac{\boldsymbol{p}_{i,k}\boldsymbol{\sigma}_{i,k}}{\sum_{i,k} \boldsymbol{p}_{i,k}\boldsymbol{\sigma}_{i,k}}$$

which completes the proof. □

### B.3   PROOF OF THEOREM 2

*Proof.* To prove this theorem, we need a few more lemmas.

**Lemma 5.** *Algorithm 1's computational cost is $O(LKN)$ and space cost is $O(L^2K)$. Furthermore, for any $n > 2LK$, after the $n-1$th iteration and before the $n$th iteration, we have*

$$\boldsymbol{N}_{\ell,k} = \boldsymbol{N}_{\ell,k,n} = \sum_{i=1}^{n-1} \mathbb{1}_{I_i=(\ell,k)}$$

$$\hat{\boldsymbol{\mu}}_{\ell,k,j} = \hat{\boldsymbol{\mu}}_{\ell,k,j,n} = \frac{1}{\boldsymbol{N}_{\ell,k,n}} \sum_{i=1}^{n-1} \mathbb{1}_{I_i=(\ell,k)} \mathbb{1}_{\hat{y}(x_i)=j}$$

$$\hat{\boldsymbol{\sigma}}_{\ell,k} = \hat{\boldsymbol{\sigma}}_{\ell,k,n} = 1 - \frac{1}{\boldsymbol{N}_{\ell,k,n}(\boldsymbol{N}_{\ell,k,n}-1)} \sum_{i=1}^{n-1} \sum_{j=1,j\neq i}^{n-1} \mathbb{1}_{I_i=I_j=(\ell,k)} \mathbb{1}_{\hat{y}(x_i)=\hat{y}(x_j)}$$

$$\boldsymbol{H}_{\ell,k,j} = \boldsymbol{H}_{\ell,k,j,n} = \sum_{i=1}^{n-1} \mathbb{1}_{I_i=(\ell,k)} \mathbb{1}_{\hat{y}(x_i)=j}$$

*Proof.* The computational and space cost can be easily verified: as shown in Algorithm 1, the variables $\hat{\boldsymbol{\sigma}}, \hat{\boldsymbol{\mu}}, \boldsymbol{H}, \boldsymbol{N}$ take space $LK, L2K, L^2K, LK$. Therefore, the space is bounded by $O(L^2K)$. For the first $2LK$ iterations (line 3-8) in Algorithm 1, the computation cost is clearly $O(LK)$.

For the rest iterations (line 10- 16), the most expensive cost is computing $I_n$, which requires $LK$ computations per iteration. Therefore, the total computational cost is $O(LKN)$.

Next we show that the above four equations hold for every $n > 2LK$. We prove this by induction.

1) $n = 2LK + 1$: One can easily verify this by plugging the initial values established in line 3-8 in Algorithm 1.

2) Suppose the four equations hold for the case when $n = m$. Now consider $n = m + 1$. Now let us consider two cases.

- Any $\ell, k$ such that $I_{m+1} \neq (\ell, k)$: There is nothing update,

$$\boldsymbol{N}_{\ell,k,m+1} = \boldsymbol{N}_{\ell,k,m} = \sum_{i=1}^{m-1} \mathbb{1}_{I_i=(\ell,k)} = \sum_{i=1}^{m-1} \mathbb{1}_{I_i=(\ell,k)} + 0 = \sum_{i=1}^{m-1} \mathbb{1}_{I_i=(\ell,k)} + \mathbb{1}_{I_m=(\ell,k)}$$

$$= \sum_{i=1}^{m} \mathbb{1}_{I_i=(\ell,k)}$$

Similarly, one can show that

$$\hat{\boldsymbol{\mu}}_{\ell,k,j,m+1} = \hat{\boldsymbol{\mu}}_{\ell,k,j,m} = \frac{1}{\boldsymbol{N}_{\ell,k,m}} \sum_{i=1}^{m} \mathbb{1}_{I_i=(\ell,k)} \mathbb{1}_{\hat{y}(x_i)=j}$$

$$\hat{\boldsymbol{\sigma}}_{\ell,k,m+1} = \hat{\boldsymbol{\sigma}}_{\ell,k,m} = 1 - \frac{1}{\boldsymbol{N}_{\ell,k,m}(\boldsymbol{N}_{\ell,k,m}-1)} \sum_{i=1}^{m} \sum_{j=1,j\neq i}^{m} \mathbb{1}_{I_i=I_j=(\ell,k)} \mathbb{1}_{\hat{y}(x_i)=\hat{y}(x_j)}$$

$$\boldsymbol{H}_{\ell,k,j,m+1} = \boldsymbol{H}_{\ell,k,j,m} = \sum_{i=1}^{m} \mathbb{1}_{I_i=(\ell,k)} \mathbb{1}_{\hat{y}(x_i)=j}$$

- For some $\ell^*, k^*$ such that $I_{m+1} = (\ell^*, k^*)$.

  Let us first consider $\boldsymbol{N}_{\ell^*,k^*}$. We increment $\boldsymbol{N}_{\ell^*,k^*}$ by one, and thus

$$\boldsymbol{N}_{\ell^*,k^*,m+1} = \boldsymbol{N}_{\ell^*,k^*,m} + 1 = \sum_{i=1}^{m-1} \mathbb{1}_{I_i=(\ell^*,{}^*k)} + 1 = \sum_{i=1}^{m-1} \mathbb{1}_{I_i=(\ell^*,k^*)} + \mathbb{1}_{I_m=(\ell^*,k^*)}$$

$$= \sum_{i=1}^{m} \mathbb{1}_{I_i=(\ell^*,k^*)}$$

  Next we consider $\hat{\boldsymbol{\mu}}_{\ell^*,k^*,j}$. Using a similar argument as above, we have

$$
\begin{aligned}
\hat{\boldsymbol{\mu}}_{\ell^*,k^*,j,m+1} &= \hat{\boldsymbol{\mu}}_{\ell^*,k^*,j,m} + \frac{\mathbb{1}_{\hat{y}(x_m)=j} - \hat{\boldsymbol{\mu}}_{\ell^*,k^*,j,m}}{\boldsymbol{N}_{\ell^*,k^*,m+1}}, \\
&= \frac{\boldsymbol{N}_{\ell^*,k^*,m+1} - 1}{\boldsymbol{N}_{\ell^*,k^*,m+1}} \hat{\boldsymbol{\mu}}_{\ell^*,k^*,j,m} + \frac{\mathbb{1}_{\hat{y}(x_m)=j}}{\boldsymbol{N}_{\ell^*,k^*,m+1}}, \\
&= \frac{\boldsymbol{N}_{\ell^*,k^*,m}}{\boldsymbol{N}_{\ell^*,k^*,m+1}} \hat{\boldsymbol{\mu}}_{\ell^*,k^*,j,m} + \frac{\mathbb{1}_{\hat{y}(x_m)=j}}{\boldsymbol{N}_{\ell^*,k^*,m+1}}, \\
&= \frac{\boldsymbol{N}_{\ell^*,k^*,m}}{\boldsymbol{N}_{\ell^*,k^*,m+1}} \frac{1}{\boldsymbol{N}_{\ell^*,k^*,m}} \sum_{i=1}^{m-1} \mathbb{1}_{I_i=(\ell^*,k^*)} \mathbb{1}_{\hat{y}(x_i)=j} + \frac{\mathbb{1}_{\hat{y}(x_m)=j}}{\boldsymbol{N}_{\ell^*,k^*,m+1}} \\
&= \frac{1}{\boldsymbol{N}_{\ell^*,k^*,m+1}} \sum_{i=1}^{m} \mathbb{1}_{I_i=(\ell^*,k^*)} \mathbb{1}_{\hat{y}(x_i)=j}
\end{aligned}
$$

where the first equation is due to the update rule of $\hat{\boldsymbol{\mu}}_{\ell^*,k^*,j}$, the second equation is simply grouping by $\hat{\boldsymbol{\mu}}_{\ell^*,k^*,j,m}$, the third equation is due to the fact that $\boldsymbol{N}_{\ell^*,k^*,m+1} = \boldsymbol{N}_{\ell^*,k^*,m}+1$, the forth equation is due to the induction assumption, and the forth equation is simply algebraic rewriting.

Now let us consider $\hat{\boldsymbol{\sigma}}^2_{\ell^*,k^*,m+1}$. We can write

$$
\begin{aligned}
&\hat{\boldsymbol{\sigma}}^2_{\ell^*,k^*,m+1}\\
=&\hat{\boldsymbol{\sigma}}^2_{\ell^*,k^*,m} + \frac{2}{\boldsymbol{N}_{\ell^*,k^*,m+1}}(1 - \frac{\boldsymbol{H}_{\ell^*,k^*,\hat{y}(x_m),m}}{\boldsymbol{N}_{\ell^*,k^*,m+1} - 1} - \hat{\boldsymbol{\sigma}}^2_{\ell^*,k^*,m})\\
=&\frac{\boldsymbol{N}_{\ell^*,k^*,m+1} - 2}{\boldsymbol{N}_{\ell^*,k^*,m+1}}\hat{\boldsymbol{\sigma}}^2_{\ell^*,k^*,m} + \frac{2}{\boldsymbol{N}_{\ell^*,k^*,m+1}}(1 - \frac{\boldsymbol{H}_{\ell^*,k^*,\hat{y}(x_m),m}}{\boldsymbol{N}_{\ell^*,k^*,m+1} - 1})\\
=&\frac{\boldsymbol{N}_{\ell^*,k^*,m+1} - 2}{\boldsymbol{N}_{\ell^*,k^*,m+1}}(1 - \frac{1}{\boldsymbol{N}_{\ell^*,k^*,m}(\boldsymbol{N}_{\ell^*,k^*,m} - 1)} \sum_{i=1}^{m-1}\sum_{j=1,j\neq i}^{m-1} \mathbb{1}_{I_i=I_j=(\ell^*,k^*)}\mathbb{1}_{\hat{y}(x_i)=\hat{y}(x_j)})\\
&+\frac{2}{\boldsymbol{N}_{\ell^*,k^*,m+1}}(1 - \frac{\boldsymbol{H}_{\ell^*,k^*,\hat{y}(x_m),m}}{\boldsymbol{N}_{\ell^*,k^*,m+1} - 1})\\
=&\frac{\boldsymbol{N}_{\ell^*,k^*,m+1} - 2}{\boldsymbol{N}_{\ell^*,k^*,m+1}}(1 - \frac{1}{(\boldsymbol{N}_{\ell^*,k^*,m+1} - 2)(\boldsymbol{N}_{\ell^*,k^*,m+1} - 1)} \sum_{i=1}^{m-1}\sum_{j=1,j\neq i}^{m-1} \mathbb{1}_{I_i=I_j=(\ell^*,k^*)}\mathbb{1}_{\hat{y}(x_i)=\hat{y}(x_j)})\\
&+\frac{2}{\boldsymbol{N}_{\ell^*,k^*,m+1}}(1 - \frac{\boldsymbol{H}_{\ell^*,k^*,\hat{y}(x_m),m}}{\boldsymbol{N}_{\ell^*,k^*,m+1} - 1})\\
=&1 - \frac{1}{\boldsymbol{N}_{\ell^*,k^*,m+1}(\boldsymbol{N}_{\ell^*,k^*,m+1} - 1)} \sum_{i=1}^{m-1}\sum_{j=1,j\neq i}^{m-1} \mathbb{1}_{I_i=I_j=(\ell^*,k^*)}\mathbb{1}_{\hat{y}(x_i)=\hat{y}(x_j)}\\
&-\frac{2\boldsymbol{H}_{\ell^*,k^*,\hat{y}(x_m),m}}{\boldsymbol{N}_{\ell^*,k^*,m+1}(\boldsymbol{N}_{\ell^*,k^*,m+1} - 1)}
\end{aligned}
$$

where the first equation is by the update rule in Algorithm 1, the second equation is simply rearranging the terms, the third equation uses the induction assumption, the forth one uses the update rule on $\boldsymbol{N}_{\ell^*,k^*}$ and thus $\boldsymbol{N}_{\ell^*,k^*,m} = \boldsymbol{N}_{\ell^*,k^*,m+1} - 1$, and the fifth equation is also rearranging the terms.

On the other hand, by induction assumption, we have

$$
\boldsymbol{H}_{\ell^*,k^*,\hat{y}(x_m),m} = \sum_{i=1}^{m-1} \mathbb{1}_{I_i=(\ell^*,k^*)}\mathbb{1}_{\hat{y}(x_i)=\hat{y}(x_m)}
$$

And thus

$$
\begin{aligned}
&\sum_{i=1}^{m-1}\sum_{j=1,j\neq i}^{m-1} \mathbb{1}_{I_i=I_j=(\ell^*,k^*)}\mathbb{1}_{\hat{y}(x_i)=\hat{y}(x_j)} + 2\boldsymbol{H}_{\ell^*,k^*,\hat{y}(x_m),m}\\
=&\sum_{i=1}^{m}\sum_{j=1,j\neq i}^{m} \mathbb{1}_{I_i=(\ell^*,k^*)}\mathbb{1}_{\hat{y}(x_i)=\hat{y}(x_m)}
\end{aligned}
$$

Hence, the above equation becomes

$$
\hat{\boldsymbol{\sigma}}^2_{\ell^*,k^*,m+1} = 1 - \frac{1}{\boldsymbol{N}_{\ell^*,k^*,m+1}(\boldsymbol{N}_{\ell^*,k^*,m+1} - 1)} \sum_{i=1}^{m}\sum_{j=1,j\neq i}^{m} \mathbb{1}_{I_i=I_j=(\ell^*,k^*)}\mathbb{1}_{\hat{y}(x_i)=\hat{y}(x_j)}
$$

Finally, let us consider $\boldsymbol{H}_{\ell^*,k^*,j}$. If $j \neq \hat{y}(x_m)$, it is clear that

$$
\begin{aligned}
\boldsymbol{H}_{\ell^*,k^*,j,m+1} =\boldsymbol{H}_{\ell^*,k^*,j,m} &= \sum_{i=1}^{m-1} \mathbb{1}_{I_i=(\ell^*,k^*)} \mathbb{1}_{\hat{y}(x_i)=j} + 0 \\
&= \sum_{i=1}^{m-1} \mathbb{1}_{I_i=(\ell^*,k^*)} \mathbb{1}_{\hat{y}(x_i)=j} + \mathbb{1}_{I_i=(\ell^*,k^*)} \mathbb{1}_{\hat{y}(x_m)=j} \\
&= \sum_{i=1}^{m} \mathbb{1}_{I_i=(\ell^*,k^*)} \mathbb{1}_{\hat{y}(x_i)=j}
\end{aligned}
$$

where the first is due to that there is no update for this $j$, the third equation is due to the fact that $\hat{y}(x_m) \neq j$, and all the other equations are algebraic rewriting.

If $j = \hat{y}(x_m)$, it is clear that

$$
\begin{aligned}
\boldsymbol{H}_{\ell^*,k^*,j,m+1} =\boldsymbol{H}_{\ell^*,k^*,j,m} + 1 &= \sum_{i=1}^{m-1} \mathbb{1}_{I_i=(\ell^*,k^*)} \mathbb{1}_{\hat{y}(x_i)=j} + 1 \\
&= \sum_{i=1}^{m-1} \mathbb{1}_{I_i=(\ell^*,k^*)} \mathbb{1}_{\hat{y}(x_i)=j} + \mathbb{1}_{I_i=(\ell^*,k^*)} \mathbb{1}_{\hat{y}(x_m)=j} \\
&= \sum_{i=1}^{m} \mathbb{1}_{I_i=(\ell^*,k^*)} \mathbb{1}_{\hat{y}(x_i)=j}
\end{aligned}
$$

where the first is due to that there is no update for this $j$, the third equation is due to the fact that $\hat{y}(x_m) = j$, and all the other equations are algebraic rewriting.

That is to say, we have shown that,

$$
\boldsymbol{N}_{\ell,k,m+1} = \sum_{i=1}^{m} \mathbb{1}_{I_i=(\ell,k)}
$$

$$
\hat{\boldsymbol{\mu}}_{\ell,k,j,m+1} = \frac{1}{\boldsymbol{N}_{\ell,k,m+1}} \sum_{i=1}^{m} \mathbb{1}_{I_i=(\ell,k)} \mathbb{1}_{\hat{y}(x_i)=j}
$$

$$
\hat{\boldsymbol{\sigma}}_{\ell,k,m+1} = 1 - \frac{1}{\boldsymbol{N}_{\ell,k,m+1}(\boldsymbol{N}_{\ell,k,m+1} - 1)} \sum_{i=1}^{m} \sum_{j=1,j\neq i}^{m} \mathbb{1}_{I_i=I_j=(\ell,k)} \mathbb{1}_{\hat{y}(x_i)=\hat{y}(x_j)}
$$

$$
\boldsymbol{H}_{\ell,k,j,m+1} = \sum_{i=1}^{m} \mathbb{1}_{I_i=(\ell,k)} \mathbb{1}_{\hat{y}(x_i)=j}
$$

always hold. By induction, we can say that for any $n > 2LK$, the original equations hold, which completes the proof. □

**Lemma 6.** *Suppose that the event A holds. Set $\delta = 2e^{-a}$. Then for each $\ell, k$, we have*

$$
\frac{1}{\boldsymbol{N}_{\ell,k}} \leq \frac{1}{\boldsymbol{N}_{\ell,k}^*} \left[ 1 + 4LKN^{-1} + \frac{4}{\boldsymbol{\sigma}_{\min}} \sqrt[4]{\frac{\log 2/\delta}{\Delta_{\min}}} N^{-\frac{1}{4}} \right]
$$

*for any $1 \leq \ell \leq L, 1 \leq k \leq K$.*

*Proof.* To show this, let us first establish the following useful lemma.

**Lemma 7.** *Suppose that the event A holds. If Algorithm 1 draws at least one sample from $D_{\ell_0,k_0}$ after the first $2LK$ iterations, we must have, for every $\ell, k$,*

$$
\boldsymbol{N}_{\ell,k} \geq (\boldsymbol{N}_{\ell_0,k_0} - 1) \boldsymbol{\sigma}_{\ell,k} \frac{\boldsymbol{p}_{\ell,k}}{\boldsymbol{p}_{\ell_0,k_0}} \left( \boldsymbol{\sigma}_{\ell_0,k_0} + 2 \sqrt[4]{\frac{\log 2/\delta}{2(\boldsymbol{N}_{\ell_0,k_0} - `1)}} \right)^{-1}
$$

*Proof.* Since the event $A$ holds, we have

$$\left\{\left|\sqrt{1-\frac{1}{t(t-1)}\sum_{i=1}^{t}\sum_{j=1,j\neq i}^{t}\mathbb{1}_{\boldsymbol{z}_{\ell,k,i}=\boldsymbol{z}_{\ell,k,j}}}-\boldsymbol{\sigma}_{\ell,k}\right|\leq\sqrt[4]{\frac{\log 2/\delta}{2t}}\right\}$$

for every $\ell, k, t$. Since this holds for every fixed $t$, it should also holds for any random variable $t$. Specifically, we must have

$$\left|\sqrt{1-\frac{1}{\boldsymbol{N}_{\ell,k,n}(\boldsymbol{N}_{\ell,k,n}-1)}\sum_{i=1}^{\boldsymbol{N}_{\ell,k,n}}\sum_{j=1,j\neq i}^{\boldsymbol{N}_{\ell,k,n}}\mathbb{1}_{\boldsymbol{z}_{\ell,k,i}=\boldsymbol{z}_{\ell,k,j}}}-\boldsymbol{\sigma}_{\ell,k}\right|\leq\sqrt[4]{\frac{\log 2/\delta}{2\boldsymbol{N}_{\ell,k,n}}}$$

Note that, by definition,

$$\hat{\boldsymbol{\sigma}}_{\ell,k,n}=\sqrt{1-\frac{1}{\boldsymbol{N}_{\ell,k,n}(\boldsymbol{N}_{\ell,k,n}-1)}\sum_{i=1}^{\boldsymbol{N}_{\ell,k,n}}\sum_{j=1,j\neq i}^{\boldsymbol{N}_{\ell,k,n}}\mathbb{1}_{\boldsymbol{z}_{\ell,k,i}=\boldsymbol{z}_{\ell,k,j}}}$$

We can then rewrite the above inequality as

$$|\hat{\boldsymbol{\sigma}}_{\ell,k,n}-\boldsymbol{\sigma}_{\ell,k}|\leq\sqrt[4]{\frac{\log 2/\delta}{2\boldsymbol{N}_{\ell,k,n}}}$$

That is to say,

$$\boldsymbol{\sigma}_{\ell,k}-\sqrt[4]{\frac{\log 2/\delta}{2\boldsymbol{N}_{\ell,k,n}}}\leq\hat{\boldsymbol{\sigma}}_{\ell,k,n}\leq\boldsymbol{\sigma}_{\ell,k}+\sqrt[4]{\frac{\log 2/\delta}{2\boldsymbol{N}_{\ell,k,n}}}$$

Adding $\sqrt[4]{\frac{\log 2/\delta}{2\boldsymbol{N}_{\ell,k,n}}}$ to both sides, this becomes

$$\boldsymbol{\sigma}_{\ell,k}\leq\hat{\boldsymbol{\sigma}}_{\ell,k,n}+\sqrt[4]{\frac{\log 2/\delta}{2\boldsymbol{N}_{\ell,k,n}}}\leq\boldsymbol{\sigma}_{\ell,k}+2\sqrt[4]{\frac{\log 2/\delta}{2\boldsymbol{N}_{\ell,k,n}}}$$

Multiplying both sides by $\frac{\boldsymbol{p}_{\ell,k}}{\boldsymbol{N}_{\ell,k,n}}$, we have

$$\frac{\boldsymbol{p}_{\ell,k}}{\boldsymbol{N}_{\ell,k,n}}\boldsymbol{\sigma}_{\ell,k}\leq\frac{\boldsymbol{p}_{\ell,k}}{\boldsymbol{N}_{\ell,k,n}}\left(\hat{\boldsymbol{\sigma}}_{\ell,k,n}+\sqrt[4]{\frac{\log 2/\delta}{2\boldsymbol{N}_{\ell,k,n}}}\right)\leq\frac{\boldsymbol{p}_{\ell,k}}{\boldsymbol{N}_{\ell,k,n}}\left(\boldsymbol{\sigma}_{\ell,k}+2\sqrt[4]{\frac{\log 2/\delta}{2\boldsymbol{N}_{\ell,k,n}}}\right) \qquad \text{(B.1)}$$

which holds for any $\ell, k, n$. Note that $N > 2LK$, there must exist some $\ell_0, k_0$, such that Algorithm 1 draws a sample from the data partition $D_{\ell_0,k_0}$ after the first $2LK$ iterations. Suppose the last time a sample is drawn from $D_{\ell_0,k_0}$ is $n_0 > 2LK$. That is to say, $\boldsymbol{N}_{\ell_0,k_0,n_0}=\boldsymbol{N}_{\ell_0,k_0,n}-1, \forall n = n_0+1,\cdots,N$. Since Algorithm 1 chooses $\ell_0, k_0$ at iteration $n_0$, by line 11 in Algorithm 1, we have

$$\ell_0,k_0=\arg\max\frac{\boldsymbol{p}_{\ell,k}}{\boldsymbol{N}_{\ell,k,n_0}}(\hat{\boldsymbol{\sigma}}_{\ell,k,n_0}+\sqrt[4]{\frac{\log 2/\delta}{2\boldsymbol{N}_{\ell,k,n_0}}})$$

By definition of $\arg\max$, we have

$$\frac{\boldsymbol{p}_{\ell_0,k_0}}{\boldsymbol{N}_{\ell_0,k_0,n_0}}(\hat{\boldsymbol{\sigma}}_{\ell_0,k_0,n_0}+\sqrt[4]{\frac{\log 2/\delta}{2\boldsymbol{N}_{\ell_0,k_0,n_0}}})\geq\frac{\boldsymbol{p}_{\ell,k}}{\boldsymbol{N}_{\ell,k,n_0}}(\hat{\boldsymbol{\sigma}}_{\ell,k,n_0}+\sqrt[4]{\frac{\log 2/\delta}{2\boldsymbol{N}_{\ell,k,n_0}}})$$

Setting $n = n_0$ in the first half of inequality B.1, we have

$$\frac{\boldsymbol{p}_{\ell,k}}{\boldsymbol{N}_{\ell,k,n_0}}\left(\hat{\boldsymbol{\sigma}}_{\ell,k,n_0}+\sqrt[4]{\frac{\log 2/\delta}{2\boldsymbol{N}_{\ell,k,n_0}}}\right)\geq\frac{\boldsymbol{p}_{\ell,k}}{\boldsymbol{N}_{\ell,k,n_0}}\boldsymbol{\sigma}_{\ell,k,n_0}$$

Combining the above two inequalities gives

$$\frac{\boldsymbol{p}_{\ell_0,k_0}}{\boldsymbol{N}_{\ell_0,k_0,n_0}}(\hat{\boldsymbol{\sigma}}_{\ell_0,k_0,n_0}+\sqrt[4]{\frac{\log 2/\delta}{2\boldsymbol{N}_{\ell_0,k_0,n_0}}})\geq\frac{\boldsymbol{p}_{\ell,k}}{\boldsymbol{N}_{\ell,k,n_0}}\boldsymbol{\sigma}_{\ell,k}$$

Noting that by definition, $\boldsymbol{N}_{\ell,k,n_0} \leq \boldsymbol{N}_{\ell,k,N} = \boldsymbol{N}_{\ell,k}$, we can lower bound $1/\boldsymbol{N}_{\ell,k,n_0}$ by $1/\boldsymbol{N}_{\ell,k}$, and the above inequality becomes

$$\frac{\boldsymbol{p}_{\ell_0,k_0}}{\boldsymbol{N}_{\ell_0,k_0,n_0}}(\hat{\boldsymbol{\sigma}}_{\ell_0,k_0,n_0} + \sqrt[4]{\frac{\log 2/\delta}{2\boldsymbol{N}_{\ell_0,k_0,n_0}}}) \geq \frac{\boldsymbol{p}_{\ell,k}}{\boldsymbol{N}_{\ell,k}}\boldsymbol{\sigma}_{\ell,k}$$

Now setting $n = n_0, \ell = \ell_0, k = k_0$ in the second half of inequality B.1, we have

$$\frac{\boldsymbol{p}_{\ell_0,k_0}}{\boldsymbol{N}_{\ell_0,k_0,n}}\left(\hat{\boldsymbol{\sigma}}_{\ell_0,k_0,n} + \sqrt[4]{\frac{\log 2/\delta}{2\boldsymbol{N}_{\ell_0,k_0,n_0}}}\right) \leq \frac{\boldsymbol{p}_{\ell_0,k_0}}{\boldsymbol{N}_{\ell_0,k_0,n_0}}\left(\boldsymbol{\sigma}_{\ell_0,k_0} + 2\sqrt[4]{\frac{\log 2/\delta}{2\boldsymbol{N}_{\ell_0,k_0,n_0}}}\right)$$

Combining the above two inequalities, we have

$$\frac{\boldsymbol{p}_{\ell_0,k_0}}{\boldsymbol{N}_{\ell_0,k_0,n_0}}(\boldsymbol{\sigma}_{\ell_0,k_0} + 2\sqrt[4]{\frac{\log 2/\delta}{2\boldsymbol{N}_{\ell_0,k_0,n_0}}}) \geq \frac{\boldsymbol{p}_{\ell,k}}{\boldsymbol{N}_{\ell,k}}\boldsymbol{\sigma}_{\ell,k}$$

Observe that $n_0$ is the last time a sample is drawn from partition $D_{\ell_0,k_0}$, we have $\boldsymbol{N}_{\ell_0,k_0,n_0} = \boldsymbol{N}_{\ell_0,k_0,n}-1, \forall n = n_0+1, \cdots, N$. Specifically, $\boldsymbol{N}_{\ell_0,k_0,n_0} = \boldsymbol{N}_{\ell_0,k_0,N}-1 = \boldsymbol{N}_{\ell_0,k_0}-1$. Replacing $\boldsymbol{N}_{\ell_0,k_0,n_0}$ by $\boldsymbol{N}_{\ell_0,k_0}-1$ in the above inequality, we get

$$\frac{\boldsymbol{p}_{\ell_0,k_0}}{\boldsymbol{N}_{\ell_0,k_0}-1}(\boldsymbol{\sigma}_{\ell_0,k_0} + 2\sqrt[4]{\frac{\log 2/\delta}{2(\boldsymbol{N}_{\ell_0,k_0}-`1)}}) \geq \frac{\boldsymbol{p}_{\ell,k}}{\boldsymbol{N}_{\ell,k}}\boldsymbol{\sigma}_{\ell,k}$$

which holds for every $\ell, k$. Rearranging the terms completes the proof. $\qquad\square$

Now we are ready to prove the bound on $\boldsymbol{N}_{\ell,k} - \boldsymbol{N}_{\ell,k}^*$.

Let us first consider the lower bound. By definition, we have

$$\sum_{\ell=1}^{L}\sum_{k=1}^{K}\boldsymbol{N}_{\ell,k} = N$$

Subtracting 2 from each element, we have

$$\sum_{\ell=1}^{L}\sum_{k=1}^{K}(\boldsymbol{N}_{\ell,k} - 2) = N - 2LK = \frac{N - 2LK}{N}N$$

Note that by definition, $N = \sum_{\ell=1}^{L}\sum_{k=1}^{K}\boldsymbol{N}_{\ell,k}^*$. We can now replace the second $N$ in the above equality, and obtain

$$\sum_{\ell=1}^{L}\sum_{k=1}^{K}(\boldsymbol{N}_{\ell,k} - 2) = \frac{N - 2LK}{N}N = \frac{N - 2LK}{N}\sum_{\ell=1}^{L}\sum_{k=1}^{K}\boldsymbol{N}_{\ell,k}^* = \sum_{\ell=1}^{L}\sum_{k=1}^{K}\frac{\boldsymbol{N}_{\ell,k}^*(N - 2LK)}{N}$$

Now let us consider two cases.

(i) Assume $\boldsymbol{N}_{\ell,k} - 2 \geq \frac{\boldsymbol{N}_{\ell,k}^*(N-2LK)}{N}$. That is to say, $\boldsymbol{N}_{\ell,k} \geq \frac{\boldsymbol{N}_{\ell,k}^*(N-2LK)}{N} + 2$. Then we have

$$\frac{1}{\boldsymbol{N}_{\ell,k}} \leq \frac{1}{\frac{\boldsymbol{N}_{\ell,k}^*(N-2LK)}{N} + 2}$$

subtracting $\frac{1}{\boldsymbol{N}^*_{\ell,k}}$ from both sides, we get

$$
\begin{aligned}
\frac{1}{\boldsymbol{N}_{\ell,k}} - \frac{1}{\boldsymbol{N}^*_{\ell,k}} &\leq \frac{1}{\frac{\boldsymbol{N}^*_{\ell,k}(N-2LK)}{N} + 2} - \frac{1}{\boldsymbol{N}^*_{\ell,k}} \\
&= \frac{\boldsymbol{N}^*_{\ell,k} - \frac{\boldsymbol{N}^*_{\ell,k}(N-2LK)}{N} - 2}{\boldsymbol{N}^*_{\ell,k} \cdot (\frac{\boldsymbol{N}^*_{\ell,k}(N-2LK)}{N} + 2)} \\
&\leq \frac{\boldsymbol{N}^*_{\ell,k} - \frac{\boldsymbol{N}^*_{\ell,k}(N-2LK)}{N}}{\boldsymbol{N}^*_{\ell,k} \cdot (\frac{\boldsymbol{N}^*_{\ell,k}(N-2LK)}{N})} \\
&= \frac{\frac{2LK\boldsymbol{N}^*_{\ell,k}}{N}}{\boldsymbol{N}^*_{\ell,k} \cdot (\frac{\boldsymbol{N}^*_{\ell,k}(N-2LK)}{N})} \\
&= \frac{2LK}{\boldsymbol{N}^*_{\ell,k}(N-2LK)}
\end{aligned}
$$

where the last inequality is simply by removing the constant 2. Now by assumption, $N > 4LK$, we have $N - 2LK < \frac{1}{2}N$. The above inequality can be further simplified as

$$
\frac{1}{\boldsymbol{N}_{\ell,k}} - \frac{1}{\boldsymbol{N}^*_{\ell,k}} \leq \frac{2LK}{\boldsymbol{N}^*_{\ell,k}(N-2LK)} \leq \frac{4LK}{\boldsymbol{N}^*_{\ell,k}N}
$$

By definition, we have $\boldsymbol{N}^*_{\ell,k} = N\Delta_{\ell,k} \leq N\Delta_{\min}$. Therefore, we have

$$
\frac{1}{\boldsymbol{N}_{\ell,k}} - \frac{1}{\boldsymbol{N}^*_{\ell,k}} \leq \frac{2LK}{\boldsymbol{N}^*_{\ell,k}(N-2LK)} \leq \frac{4LK}{\boldsymbol{N}^*_{\ell,k}N}
$$

That is to say,

$$
\frac{1}{\boldsymbol{N}_{\ell,k}} \leq \frac{1}{\boldsymbol{N}^*_{\ell,k}}\left[1 + \frac{4LK}{N}\right]
$$

And thus, apparently,

$$
\frac{1}{\boldsymbol{N}_{\ell,k}} \leq \frac{1}{\boldsymbol{N}^*_{\ell,k}}\left[1 + 4LKN^{-1} + \frac{4}{\boldsymbol{\sigma}_{\min}}\sqrt[4]{\frac{\log 2/\delta}{\Delta_{\min}}}N^{-\frac{1}{4}}\right]
$$

(ii) Assume $\boldsymbol{N}_{\ell,k} - 2 < \frac{\boldsymbol{N}^*_{\ell,k}(N-2LK)}{N}$. Then there must exists some $\ell_0, k_0$ such that $\boldsymbol{N}_{\ell_0,k_0} - 2 > \frac{\boldsymbol{N}^*_{\ell_0,k_0}(N-2LK)}{N} > 0$. That is to say, Algorithm 1 draws at least one sample from $D_{\ell_0,k_0}$ after the first $2LK$ iterations. By Lemma 7, we must have

$$
\boldsymbol{N}_{\ell,k} \geq (\boldsymbol{N}_{\ell_0,k_0} - 1)\boldsymbol{\sigma}_{\ell,k}\frac{\boldsymbol{p}_{\ell,k}}{\boldsymbol{p}_{\ell_0,k_0}}\left(\boldsymbol{\sigma}_{\ell_0,k_0} + 2\sqrt[4]{\frac{\log 2/\delta}{2(\boldsymbol{N}_{\ell_0,k_0} - `1)}}\right)^{-1}
$$

$\boldsymbol{N}_{\ell_0,k_0} - 2 > \frac{\boldsymbol{N}^*_{\ell_0,k_0}(N-2LK)}{N}$ implies

$$
\boldsymbol{N}_{\ell_0,k_0} - 1 > \boldsymbol{N}_{\ell_0,k_0-2} > \frac{\boldsymbol{N}^*_{\ell_0,k_0}(N-2LK)}{N}
$$

Therefore, we can use this lower bound on $\boldsymbol{N}_{\ell_0,k_0} - 1$ in the above inequality and obtain

$$\boldsymbol{N}_{\ell,k} \geq \frac{\boldsymbol{N}^*_{\ell_0,k_0}(N-2LK)}{N}\boldsymbol{\sigma}_{\ell,k}\frac{\boldsymbol{p}_{\ell,k}}{\boldsymbol{p}_{\ell_0,k_0}}\left(\boldsymbol{\sigma}_{\ell_0,k_0} + 2\sqrt[4]{\frac{\log 2/\delta}{2\frac{\boldsymbol{N}^*_{\ell_0,k_0}(N-2LK)}{N}}}\right)^{-1}$$

$$=\frac{\boldsymbol{N}^*_{\ell_0,k_0}(N-2LK)}{N}\frac{\boldsymbol{\sigma}_{\ell,k}\boldsymbol{p}_{\ell,k}}{\boldsymbol{\sigma}_{\ell_0,k_0}\boldsymbol{p}_{\ell_0,k_0}}\left(1 + \frac{2}{\boldsymbol{\sigma}_{\ell_0,k_0}}\sqrt[4]{\frac{\log 2/\delta}{2\frac{\boldsymbol{N}^*_{\ell_0,k_0}(N-2LK)}{N}}}\right)^{-1}$$

$$=\frac{\boldsymbol{N}^*_{\ell,k}(N-2LK)}{N}\left(1 + \frac{2}{\boldsymbol{\sigma}_{\ell_0,k_0}}\sqrt[4]{\frac{\log 2/\delta}{2\frac{\boldsymbol{N}^*_{\ell_0,k_0}(N-2LK)}{N}}}\right)^{-1}$$

where the first equality is by dividing $\boldsymbol{\sigma}_{\ell_0,k_0}$ at both denominator and numerator, and the second equality uses the fact that $\boldsymbol{N}^*_{\ell,k}$ is proportional to $p_{\ell,k}\boldsymbol{\sigma}_{\ell,k}$. Taking inverse of the above inequality gives

$$\frac{1}{\boldsymbol{N}_{\ell,k}} \leq \frac{N}{\boldsymbol{N}^*_{\ell,k}(N-2LK)}\left(1 + \frac{2}{\boldsymbol{\sigma}_{\ell_0,k_0}}\sqrt[4]{\frac{\log 2/\delta}{2\frac{\boldsymbol{N}^*_{\ell_0,k_0}(N-2LK)}{N}}}\right)$$

Now let us simplify this inequality. Let us first expand all terms and obtain

$$\frac{1}{\boldsymbol{N}_{\ell,k}} \leq \frac{N}{\boldsymbol{N}^*_{\ell,k}(N-2LK)}\left(1 + \frac{2}{\boldsymbol{\sigma}_{\ell_0,k_0}}\sqrt[4]{\frac{\log 2/\delta}{2\frac{\boldsymbol{N}^*_{\ell_0,k_0}(N-2LK)}{N}}}\right)$$

$$= \frac{1}{\boldsymbol{N}^*_{\ell,k}} + \frac{2LK}{\boldsymbol{N}^*_{\ell,k}(N-2LK)} + \frac{N}{\boldsymbol{N}^*_{\ell,k}(N-2LK)}\cdot\frac{2}{\boldsymbol{\sigma}_{\ell_0,k_0}}\sqrt[4]{\frac{\log 2/\delta}{2\frac{\boldsymbol{N}^*_{\ell_0,k_0}(N-2LK)}{N}}}$$

$$= \frac{1}{\boldsymbol{N}^*_{\ell,k}} + \frac{2LK}{\boldsymbol{N}^*_{\ell,k}(N-2LK)} + \frac{2}{\boldsymbol{\sigma}_{\ell_0,k_0}}\sqrt[4]{\left(\frac{N}{N-2LK}\right)^5\frac{\log 2/\delta}{2\boldsymbol{N}^{*4}_{\ell,k}\boldsymbol{N}^*_{\ell_0,k_0}}}$$

For the second term, by assumption, $N > 4LK$ and thus $N - 2LK > 1/2N$, we have

$$\frac{2LK}{N-2LK} \leq \frac{4LK}{N}$$

Thus the above equation becomes

$$\frac{1}{\boldsymbol{N}_{\ell,k}} \leq \frac{1}{\boldsymbol{N}^*_{\ell,k}} + \frac{4LK}{\boldsymbol{N}^*_{\ell,k}N} + \frac{2}{\boldsymbol{\sigma}_{\ell_0,k_0}}\sqrt[4]{\left(\frac{N}{N-2LK}\right)^5\frac{\log 2/\delta}{2\boldsymbol{N}^{*4}_{\ell,k}\boldsymbol{N}^*_{\ell_0,k_0}}}$$

For the third term, $N > 4LK$ also implies

$$\frac{N}{N-2LK} = 1 + \frac{2LK}{N-2LK} < 1 + \frac{2LK}{4LK-2LK} = 2$$

Thus the above inequality can be further simplified as

$$\frac{1}{\boldsymbol{N}_{\ell,k}} \leq \frac{1}{\boldsymbol{N}^*_{\ell,k}} + \frac{4LK}{\boldsymbol{N}^*_{\ell,k}N} + \frac{4}{\boldsymbol{\sigma}_{\ell_0,k_0}}\sqrt[4]{\frac{\log 2/\delta}{\boldsymbol{N}^{*4}_{\ell,k}\boldsymbol{N}^*_{\ell_0,k_0}}}$$

$$\leq \frac{1}{\boldsymbol{N}^*_{\ell,k}}\left[1 + \frac{4LK}{N} + \frac{4}{\boldsymbol{\sigma}_{\ell_0,k_0}}\sqrt[4]{\frac{\log 2/\delta}{\boldsymbol{N}^*_{\ell_0,k_0}}}\right]$$

Now by definition, $\boldsymbol{\sigma}_{\ell_0,k_0} \geq \boldsymbol{\sigma}_{\min}$, and $\boldsymbol{N}^*_{\ell_0,k_0} = N\Delta_{\ell_0,k_0} \geq N\Delta_{\min}$, we can further simplify the above inequality

$$\frac{1}{\boldsymbol{N}_{\ell,k}} \leq \frac{1}{\boldsymbol{N}^*_{\ell,k}}\left[1 + \frac{4LK}{N} + \frac{4}{\boldsymbol{\sigma}_{\ell_0,k_0}}\sqrt[4]{\frac{\log 2/\delta}{\boldsymbol{N}^*_{\ell_0,k_0}}}\right]$$

$$\leq \frac{1}{\boldsymbol{N}^*_{\ell,k}}\left[1 + \frac{4LK}{N} + \frac{4}{\boldsymbol{\sigma}_{\ell_0,k_0}}\sqrt[4]{\frac{\log 2/\delta}{N\Delta_{\min}}}\right]$$

$$\leq \frac{1}{\boldsymbol{N}^*_{\ell,k}}\left[1 + \frac{4LK}{N} + \frac{4}{\boldsymbol{\sigma}_{\min}}\sqrt[4]{\frac{\log 2/\delta}{N\Delta_{\min}}}\right]$$

That is to say,

$$\frac{1}{\boldsymbol{N}_{\ell,k}} \leq \frac{1}{\boldsymbol{N}^*_{\ell,k}}\left[1 + 4LKN^{-1} + \frac{4}{\boldsymbol{\sigma}_{\min}}\sqrt[4]{\frac{\log 2/\delta}{\Delta_{\min}}}N^{-\frac{1}{4}}\right]$$

That is to say, no matter $\boldsymbol{N}_{\ell,k} - 2 < \frac{\boldsymbol{N}^*_{\ell,k}(N-2LK)}{N}$ or not, this inequality always holds, which completes the proof. $\square$

Now we are ready to prove Theorem 2. Let us first note that the loss can be written as

$$\mathcal{L}_N = \sum_{\ell=1}^{L}\sum_{k=1}^{K}\sum_{j=1}^{L}\boldsymbol{p}_{\ell,k}^2\mathbb{E}[\boldsymbol{\mu}_{\ell,k,j} - \hat{\boldsymbol{\mu}}_{\ell,k,j}]^2$$

$$= \sum_{\ell=1}^{L}\sum_{k=1}^{K}\sum_{j=1}^{L}\boldsymbol{p}_{\ell,k}^2\mathbb{E}[(\boldsymbol{\mu}_{\ell,k,j} - \frac{1}{\boldsymbol{N}_{\ell,k}}\sum_{t=1}^{\boldsymbol{N}_{\ell,k}}\mathbb{1}_{\boldsymbol{z}_{\ell,k,t}=j})^2\mathbb{1}_A] \qquad (\text{B.2})$$

$$+ \sum_{\ell=1}^{L}\sum_{k=1}^{K}\sum_{j=1}^{L}\boldsymbol{p}_{\ell,k}^2\mathbb{E}[(\boldsymbol{\mu}_{\ell,k,j} - \frac{1}{\boldsymbol{N}_{\ell,k}}\sum_{t=1}^{\boldsymbol{N}_{\ell,k}}\mathbb{1}_{\boldsymbol{z}_{\ell,k,t}=j})^2\mathbb{1}_{A^C}]$$

Let us first consider the first term.

$$\sum_{\ell=1}^{L}\sum_{k=1}^{K}\sum_{j=1}^{L}\boldsymbol{p}_{\ell,k}^2\mathbb{E}[(\boldsymbol{\mu}_{\ell,k,j} - \hat{\boldsymbol{\mu}}_{\ell,k,j})^2\mathbb{1}_A]$$

$$= \sum_{\ell=1}^{L}\sum_{k=1}^{K}\sum_{j=1}^{L}\boldsymbol{p}_{\ell,k}^2\mathbb{E}[(\boldsymbol{\mu}_{\ell,k,j} - \frac{1}{\boldsymbol{N}_{\ell,k}}\sum_{t=1}^{\boldsymbol{N}_{\ell,k}}\mathbb{1}_{\boldsymbol{z}_{\ell,k,t}=j})^2\mathbb{1}_A] \qquad (\text{B.3})$$

$$= \sum_{\ell=1}^{L}\sum_{k=1}^{K}\sum_{j=1}^{L}\boldsymbol{p}_{\ell,k}^2\mathbb{E}\left[\frac{1}{\boldsymbol{N}_{\ell,k}^2}\left(\boldsymbol{N}_{\ell,k}\boldsymbol{\mu}_{\ell,k,j} - \sum_{t=1}^{\boldsymbol{N}_{\ell,k}}\mathbb{1}_{\boldsymbol{z}_{\ell,k,t}=j}\right)^2\mathbb{1}_A\right]$$

where we plug in the definition of $\hat{\boldsymbol{\mu}}$. By Lemma 6, we have the upper bound on $1/\boldsymbol{N}_{\ell,k}$

$$\frac{1}{\boldsymbol{N}_{\ell,k}} \leq \frac{1}{\boldsymbol{N}^*_{\ell,k}}\left[1 + 4LKN^{-1} + \frac{4}{\boldsymbol{\sigma}_{\min}}\sqrt[4]{\frac{\log 2/\delta}{\Delta_{\min}}}N^{-\frac{1}{4}}\right]$$

Therefore, we can use this inequality to obtain

$$\mathbb{E}\left[\frac{1}{\boldsymbol{N}_{\ell,k}^2}\left(\boldsymbol{N}_{\ell,k}\boldsymbol{\mu}_{\ell,k,j} - \sum_{t=1}^{\boldsymbol{N}_{\ell,k}}\mathbb{1}_{\boldsymbol{z}_{\ell,k,t}=j}\right)^2\mathbb{1}_A\right]$$

$$\leq [\frac{1}{\boldsymbol{N}^*_{\ell,k}} + \frac{4LK}{\Delta_{\min}}N^{-2} + \frac{4}{\boldsymbol{\sigma}_{\min}}\sqrt[4]{\frac{\log 2/\delta}{\Delta_{\min}^5}}N^{-\frac{5}{4}}]^2\mathbb{E}\left[\left(\boldsymbol{N}_{\ell,k}\boldsymbol{\mu}_{\ell,k,j} - \sum_{t=1}^{\boldsymbol{N}_{\ell,k}}\mathbb{1}_{\boldsymbol{z}_{\ell,k,t}=j}\right)^2\mathbb{1}_A\right]$$

$$(\text{B.4})$$

It is not hard to see that $\boldsymbol{N}_{\ell,k}$ is a stopping time. In fact, for any $\ell, k$, and any time $n$, a new sample is drawn purely based on estimated uncertainty score $\hat{\boldsymbol{\sigma}}$ and observed sample number $\boldsymbol{N}_{\ell,k,n-1}$ up to the current iteration, which is part of the history. As $\boldsymbol{N}_{\ell,k} < N$ is bounded, $\boldsymbol{N}_{\ell,k}$ is a stopping time. Hence, we can apply Lemma 4, and obtain

$$\mathbb{E}\left[\left(\boldsymbol{N}_{\ell,k}\boldsymbol{\mu}_{\ell,k,j} - \sum_{t=1}^{\boldsymbol{N}_{\ell,k}} \mathbb{1}_{\boldsymbol{z}_{\ell,k,t}=j}\right)^2 \mathbb{1}_A\right] \leq \mathbb{E}\left[\left(\boldsymbol{N}_{\ell,k}\boldsymbol{\mu}_{\ell,k,j} - \sum_{t=1}^{\boldsymbol{N}_{\ell,k}} \mathbb{1}_{\boldsymbol{z}_{\ell,k,t}=j}\right)^2\right]$$
$$\leq \mathbb{E}[\boldsymbol{N}_{\ell,k}]\Pr[\boldsymbol{z}_{\ell,k,1}=j](1 - \Pr[\boldsymbol{z}_{\ell,k,1}=j])$$

where the first inequality uses the fact that square term must be non-negative, and the second inequality uses the fact that, for Bernoulli distribution with mean $a$, its variance is $a(1-a)$. Applying this in inequality B.4, we have

$$\mathbb{E}\left[\frac{1}{\boldsymbol{N}_{\ell,k}^2}\left(\boldsymbol{N}_{\ell,k}\boldsymbol{\mu}_{\ell,k,j} - \sum_{t=1}^{\boldsymbol{N}_{\ell,k}} \mathbb{1}_{\boldsymbol{z}_{\ell,k,t}=j}\right)^2 \mathbb{1}_A\right]$$
$$\leq [\frac{1}{\boldsymbol{N}_{\ell,k}^*} + \frac{4LK}{\Delta_{\min}}N^{-2} + \frac{4}{\boldsymbol{\sigma}_{\min}}\sqrt[4]{\frac{\log 2/\delta}{\Delta_{\min}^5}}N^{-\frac{5}{4}}]^2\mathbb{E}\left[\left(\boldsymbol{N}_{\ell,k}\boldsymbol{\mu}_{\ell,k,j} - \sum_{t=1}^{\boldsymbol{N}_{\ell,k}} \mathbb{1}_{\boldsymbol{z}_{\ell,k,t}=j}\right)^2 \mathbb{1}_A\right]$$
$$\leq [\frac{1}{\boldsymbol{N}_{\ell,k}^*} + \frac{4LK}{\Delta_{\min}}N^{-2} + \frac{4}{\boldsymbol{\sigma}_{\min}}\sqrt[4]{\frac{\log 2/\delta}{\Delta_{\min}^5}}N^{-\frac{5}{4}}]^2\mathbb{E}[\boldsymbol{N}_{\ell,k}]\Pr[\boldsymbol{z}_{\ell,k,1}=j](1 - \Pr[\boldsymbol{z}_{\ell,k,1}=j])$$

Now applying this in equality B.3, we get

$$\sum_{\ell=1}^{L}\sum_{k=1}^{K}\sum_{j=1}^{L}\boldsymbol{p}_{\ell,k}^2\mathbb{E}[(\boldsymbol{\mu}_{\ell,k,j} - \hat{\boldsymbol{\mu}}_{\ell,k,j})^2\mathbb{1}_A]$$
$$= \sum_{\ell=1}^{L}\sum_{k=1}^{K}\sum_{j=1}^{L}\boldsymbol{p}_{\ell,k}^2\mathbb{E}\left[\frac{1}{\boldsymbol{N}_{\ell,k}^2}\left(\boldsymbol{N}_{\ell,k}\boldsymbol{\mu}_{\ell,k,j} - \sum_{t=1}^{\boldsymbol{N}_{\ell,k}} \mathbb{1}_{\boldsymbol{z}_{\ell,k,t}=j}\right)^2 \mathbb{1}_A\right]$$
$$\leq \sum_{\ell=1}^{L}\sum_{k=1}^{K}\sum_{j=1}^{L}\boldsymbol{p}_{\ell,k}^2[\frac{1}{\boldsymbol{N}_{\ell,k}^*} + \frac{4LK}{\Delta_{\min}}N^{-2} + \frac{4}{\boldsymbol{\sigma}_{\min}}\sqrt[4]{\frac{\log 2/\delta}{\Delta_{\min}^5}}N^{-\frac{5}{4}}]^2\mathbb{E}[\boldsymbol{N}_{\ell,k}]\Pr[\boldsymbol{z}_{\ell,k,1}=j](1 - \Pr[\boldsymbol{z}_{\ell,k,1}=j])$$
$$= \sum_{\ell=1}^{L}\sum_{k=1}^{K}\boldsymbol{p}_{\ell,k}^2\boldsymbol{\sigma}_{\ell,k}^2[\frac{1}{\boldsymbol{N}_{\ell,k}^*} + \frac{4LK}{\Delta_{\min}}N^{-2} + \frac{4}{\boldsymbol{\sigma}_{\min}}\sqrt[4]{\frac{\log 2/\delta}{\Delta_{\min}^5}}N^{-\frac{5}{4}}]^2\mathbb{E}[\boldsymbol{N}_{\ell,k}]$$

$$(B.5)$$

where the last equation uses the fact that $\boldsymbol{\sigma}_{\ell,k} = 1 - \sum_{j=1}^{L}\Pr^2[\boldsymbol{z}_{\ell,k,1}=j] = \sum_{j=1}^{L}\Pr[\boldsymbol{z}_{\ell,k,1}=j](1 - \Pr[\boldsymbol{z}_{\ell,k,1}=j])$. Applying the inequality $1/(1+x) \leq 1 - x$

$$\frac{1}{\boldsymbol{N}_{\ell,k}} \leq \frac{1}{\boldsymbol{N}_{\ell,k}^*}\left[1 + 4LKN^{-1} + \frac{4}{\boldsymbol{\sigma}_{\min}}\sqrt[4]{\frac{\log 2/\delta}{\Delta_{\min}}}N^{\frac{1}{4}}\right]$$

Note that

$$\boldsymbol{p}_{\ell,k}^2\boldsymbol{\sigma}_{\ell,k}^2[\frac{1}{\boldsymbol{N}_{\ell,k}^*}\left[1+4LKN^{-1}+\frac{4}{\boldsymbol{\sigma}_{\min}}\sqrt[4]{\frac{\log 2/\delta}{\Delta_{\min}}}N^{-\frac{1}{4}}\right]]^2\mathbb{E}[\boldsymbol{N}_{\ell,k}]$$

$$=(\frac{\boldsymbol{p}_{\ell,k}\boldsymbol{\sigma}_{\ell,k}}{\boldsymbol{N}_{\ell,k}^*})^2\left[1+4LKN^{-1}+\frac{4}{\boldsymbol{\sigma}_{\min}}\sqrt[4]{\frac{\log 2/\delta}{\Delta_{\min}}}N^{-\frac{1}{4}}\right]^2\mathbb{E}[\boldsymbol{N}_{\ell,k}]$$

$$=N^{-2}(\sum_{\ell',k'}\boldsymbol{p}_{\ell',k'}\boldsymbol{\sigma}_{\ell',k'})^2\left[1+4LKN^{-1}+\frac{4}{\boldsymbol{\sigma}_{\min}}\sqrt[4]{\frac{\log 2/\delta}{\Delta_{\min}}}N^{-\frac{1}{4}}\right]^2\mathbb{E}[\boldsymbol{N}_{\ell,k}]$$

where the last equation is by definition of $\boldsymbol{N}_{\ell,k}$. Now applying this in inequality B.5, we have

$$\sum_{\ell=1}^{L}\sum_{k=1}^{K}\sum_{j=1}^{L}\boldsymbol{p}_{\ell,k}^2\mathbb{E}[(\boldsymbol{\mu}_{\ell,k,j}-\hat{\boldsymbol{\mu}}_{\ell,k,j})^2\mathbb{1}_A]$$

$$\leq\sum_{\ell=1}^{L}\sum_{k=1}^{K}\boldsymbol{p}_{\ell,k}^2\boldsymbol{\sigma}_{\ell,k}^2[\frac{1}{\boldsymbol{N}_{\ell,k}^*}+\frac{4LK}{\Delta_{\min}}N^{-2}+\frac{4}{\boldsymbol{\sigma}_{\min}}\sqrt[4]{\frac{\log 2/\delta}{\Delta_{\min}^5}}N^{-\frac{5}{4}}]^2\mathbb{E}[\boldsymbol{N}_{\ell,k}]$$

$$=\sum_{\ell=1}^{L}\sum_{k=1}^{K}N^{-2}(\sum_{\ell',k'}\boldsymbol{p}_{\ell',k'}\boldsymbol{\sigma}_{\ell',k'})^2\left[1+4LKN^{-1}+\frac{4}{\boldsymbol{\sigma}_{\min}}\sqrt[4]{\frac{\log 2/\delta}{\Delta_{\min}}}N^{-\frac{1}{4}}\right]^2\mathbb{E}[\boldsymbol{N}_{\ell,k}]$$

$$=N^{-2}(\sum_{\ell',k'}\boldsymbol{p}_{\ell',k'}\boldsymbol{\sigma}_{\ell',k'})^2\left[1+4LKN^{-1}+\frac{4}{\boldsymbol{\sigma}_{\min}}\sqrt[4]{\frac{\log 2/\delta}{\Delta_{\min}}}N^{-\frac{1}{4}}\right]^2\sum_{\ell=1}^{L}\sum_{k=1}^{K}\mathbb{E}[\boldsymbol{N}_{\ell,k}]$$

$$=N^{-2}(\sum_{\ell',k'}\boldsymbol{p}_{\ell',k'}\boldsymbol{\sigma}_{\ell',k'})^2\left[1+4LKN^{-1}+\frac{4}{\boldsymbol{\sigma}_{\min}}\sqrt[4]{\frac{\log 2/\delta}{\Delta_{\min}}}N^{-\frac{1}{4}}\right]^2 N$$

$$=N^{-1}(\sum_{\ell,k}\boldsymbol{p}_{\ell,k}\boldsymbol{\sigma}_{\ell,k})^2\left[1+4LKN^{-1}+\frac{4}{\boldsymbol{\sigma}_{\min}}\sqrt[4]{\frac{\log 2/\delta}{\Delta_{\min}}}N^{-\frac{1}{4}}\right]^2$$

(B.6)

where the second equation uses the fact that only $\mathbb{E}[\boldsymbol{N}_{\ell,k}]$ depends on $\ell, k$, the third equation uses the fact that $\sum_{\ell=1}^{L}\sum_{k=1}^{K}\boldsymbol{N}_{\ell,k}=N$ and thus $\sum_{\ell=1}^{L}\sum_{k=1}^{K}\mathbb{E}[\boldsymbol{N}_{\ell,k}]=N$. Note that $\delta=L^{-1}K^{-1}N^{-\frac{5}{4}}$, we have

$$N^{-1}(\sum_{\ell,k}\boldsymbol{p}_{\ell,k}\boldsymbol{\sigma}_{\ell,k})^2\left[1+4LKN^{-1}+\frac{4}{\boldsymbol{\sigma}_{\min}}\sqrt[4]{\frac{\log 2/\delta}{\Delta_{\min}}}N^{-\frac{1}{4}}\right]^2$$

$$=N^{-1}(\sum_{\ell,k}\boldsymbol{p}_{\ell,k}\boldsymbol{\sigma}_{\ell,k})^2\left[1+O(N^{-\frac{1}{4}}\log^{\frac{1}{4}}N)\right]$$

$$=N^{-1}(\sum_{\ell,k}\boldsymbol{p}_{\ell,k}\boldsymbol{\sigma}_{\ell,k})^2+O(N^{-\frac{5}{4}}\log^{\frac{1}{4}}N)$$

Applying this back to inequality B.6, we have

$$\sum_{\ell=1}^{L}\sum_{k=1}^{K}\sum_{j=1}^{L}\boldsymbol{p}_{\ell,k}^2\mathbb{E}[(\boldsymbol{\mu}_{\ell,k,j}-\hat{\boldsymbol{\mu}}_{\ell,k,j})^2\mathbb{1}_A]\leq N^{-1}(\sum_{\ell,k}\boldsymbol{p}_{\ell,k}\boldsymbol{\sigma}_{\ell,k})^2+O(N^{-\frac{5}{4}}\log^{\frac{1}{4}}N) \qquad (B.7)$$

Now consider the second term in equation B.2. As $\boldsymbol{\mu}$ and $\hat{\boldsymbol{\mu}}$ are within $\{0,1\}$, we have

$$(\boldsymbol{\mu}_{\ell,k,j}-\frac{1}{\boldsymbol{N}_{\ell,k}}\sum_{t=1}^{\boldsymbol{N}_{\ell,k}}\mathbb{1}_{\boldsymbol{z}_{\ell,k,t}=j})^2\in[0,1]$$

Therefore,

$$\sum_{\ell=1}^{L}\sum_{k=1}^{K}\sum_{j=1}^{L}\boldsymbol{p}_{\ell,k}^2\mathbb{E}[(\boldsymbol{\mu}_{\ell,k,j}-\frac{1}{\boldsymbol{N}_{\ell,k}}\sum_{t=1}^{\boldsymbol{N}_{\ell,k}}\mathbb{1}_{\boldsymbol{z}_{\ell,k,t}=j})^2\mathbb{1}_{A^C}]\le\sum_{\ell=1}^{L}\sum_{k=1}^{K}\sum_{j=1}^{L}\boldsymbol{p}_{\ell,k}^2\Pr[A^C]$$

By Lemma 3, the probability of $A$ is at least $1-KLN\delta$. Hence, the probability of $A^C$ is at most $KLN\delta$. Hence,

$$\sum_{\ell=1}^{L}\sum_{k=1}^{K}\sum_{j=1}^{L}\boldsymbol{p}_{\ell,k}^2\mathbb{E}[(\boldsymbol{\mu}_{\ell,k,j}-\frac{1}{\boldsymbol{N}_{\ell,k}}\sum_{t=1}^{\boldsymbol{N}_{\ell,k}}\mathbb{1}_{\boldsymbol{z}_{\ell,k,t}=j})^2\mathbb{1}_{A^C}]$$

$$\le\sum_{\ell=1}^{L}\sum_{k=1}^{K}\sum_{j=1}^{L}\boldsymbol{p}_{\ell,k}^2\Pr[A^C]$$

$$\le\sum_{\ell=1}^{L}\sum_{k=1}^{K}\sum_{j=1}^{L}\boldsymbol{p}_{\ell,k}^2 LKN\delta$$

$$\le\sum_{j=1}^{L}LKN\delta=L^2KN\delta$$

where the last inequality uses the fact that $\sum_{\ell=1}^{L}\sum_{k=1}^{K}\boldsymbol{p}_{\ell,k}^2\le 1$ since $\sum_{\ell=1}^{L}\sum_{k=1}^{K}\boldsymbol{p}_{\ell,k}=1$ and $\boldsymbol{p}_{\ell,k}\ge 0$. Since $\delta=L^{-2}K^{-1}N^{-\frac{9}{4}}$, we have

$$\sum_{\ell=1}^{L}\sum_{k=1}^{K}\sum_{j=1}^{L}\boldsymbol{p}_{\ell,k}^2\mathbb{E}[(\boldsymbol{\mu}_{\ell,k,j}-\frac{1}{\boldsymbol{N}_{\ell,k}}\sum_{t=1}^{\boldsymbol{N}_{\ell,k}}\mathbb{1}_{\boldsymbol{z}_{\ell,k,t}=j})^2\mathbb{1}_{A^C}]$$

$$\le L^2KN\delta\le N^{-\frac{5}{4}}$$

Applying this as well as inequality B.7 to the equation B.2, we have

$$\mathcal{L}_N=\sum_{\ell=1}^{L}\sum_{k=1}^{K}\sum_{j=1}^{L}\boldsymbol{p}_{\ell,k}^2\mathbb{E}[\boldsymbol{\mu}_{\ell,k,j}-\hat{\boldsymbol{\mu}}_{\ell,k,j}]^2$$

$$=\sum_{\ell=1}^{L}\sum_{k=1}^{K}\sum_{j=1}^{L}\boldsymbol{p}_{\ell,k}^2\mathbb{E}[(\boldsymbol{\mu}_{\ell,k,j}-\frac{1}{\boldsymbol{N}_{\ell,k}}\sum_{t=1}^{\boldsymbol{N}_{\ell,k}}\mathbb{1}_{\boldsymbol{z}_{\ell,k,t}=j})^2\mathbb{1}_A]$$

$$+\sum_{\ell=1}^{L}\sum_{k=1}^{K}\sum_{j=1}^{L}\boldsymbol{p}_{\ell,k}^2\mathbb{E}[(\boldsymbol{\mu}_{\ell,k,j}-\frac{1}{\boldsymbol{N}_{\ell,k}}\sum_{t=1}^{\boldsymbol{N}_{\ell,k}}\mathbb{1}_{\boldsymbol{z}_{\ell,k,t}=j})^2\mathbb{1}_{A^C}]$$

$$\le N^{-1}(\sum_{\ell,k}\boldsymbol{p}_{\ell,k}\boldsymbol{\sigma}_{\ell,k})^2+O(N^{-\frac{5}{4}}\log^{\frac{1}{4}}N)+N^{-\frac{5}{4}}$$

Note that the loss of the optimal allocation is simply $\mathcal{L}_N^*=N^{-1}(\sum_{\ell,k}\boldsymbol{p}_{\ell,k}\boldsymbol{\sigma}_{\ell,k})^2$. The above inequality is simply

$$\mathcal{L}_N-\mathcal{L}_N^*\le O(N^{-\frac{5}{4}}\log^{\frac{1}{4}}N)$$

which completes the proof. □

## C  EXPERIMENTAL DETAILS

**Experimental Setups.**  All experiments were run on a machine with 2 E5-2690 v4 CPUs, 160 GB RAM and 500 GB disk with Ubuntu 18.04 LTS as the OS. Our code is implemented and tested in python 3.7. All experimental results were averaged over 1500 runs, except the case study. Overall the experiments took about two month, including debugging and evaluation on all datasets. Running MASA once to draw a few thousand samples typically only takes a few seconds. Our implementation is purely in Python for demonstration purposes, and more code optimization (e.g., using cython or multi-thread) can generate a much faster implementation.

Table 2: Dataset statistics.

| Dataset | Size | # Classes | Dataset | Size | # Classes | Tasks |
|---------|------|-----------|---------|------|-----------|-------|
| FER+ | 6358 | 7 | RAFDB (Li et al.) | 15339 | 7 | *FER* |
| EXPW | 31510 | 7 | AFFECTNET | 87401 | 7 | |
| YELP | 20000 | 2 | SHOP | 62774 | 2 | *SA* |
| IMDB | 25000 | 2 | WAIMAI | 11987 | 2 | |
| DIGIT | 2000 | 10 | AUDIOMNIST | 30000 | 10 | *STT* |
| FLUENT | 30043 | 31 | COMMAND | 64727 | 31 | |

Table 3: ML services used for each task. Price unit: USD/10,000 queries. We consider three tasks, sentiment analysis (SA), facial emotion recognition (FER), and spoken command recognition (SCR).

| Tasks | ML service | Price | ML service | Price | ML service | Price |
|-------|-----------|-------|-----------|-------|-----------|-------|
| *SA* | Google NLP (GoN) | 2.5 | AMZN Comp (Ama) | 0.75 | Baidu NLP (Bai) | 3.5 |
| *FER* | Google Vision (Goo, a) | 15 | MS Face (Mic, a) | 10 | Face++ (Fac) | 5 |
| *SCR* | Google Speech (Goo, b) | 60 | MS Speech (Mic, b) | 41 | IBM Speech (IBM) | 25 |

**ML APIs and Dataset Statistics.** We focus on three common classification tasks, namely, sentiment analysis, facial emotion recognition, and spoken command recognition. For each of the tasks, we evaluated three APIs' performance in spring 2020 and spring 2021, respectively, for four datasets. The details of datasets and ML APIs are summarized in Table 2 and Table 3 respectively. Now we give more context of the datasets.

For sentiment analysis, we use four datasets, YELP, IMDB, SHOP, and WAIMAI. YELP and IMDB are both English text datasets. YELP (Dat, c) is generated by drawn twenty thousand samples from the large YELP review challenge dataset. Each original review is labeled by rating in {1,2,3,4,5}. We generate the binary label by transforming rating 1 and 2 into negative, and rating 4 and 5 into positive. Ten thousand positive reviews and ten thousand negative reviews are then randomly drawn, respectively. IMDB (Maas et al.) is a polarized sentiment analysis dataset with provided training and testing partitions. We use its testing partition which has twenty-five thousand text paragraphs. SHOP (Dat, a) and WAIMAI (Dat, b) are two Chinese text datasets. SHOP contains polarized labels for reviews for various purchases including fruits, hotels, computers. WAIMAI is a dataset for polarized delivery reviews. Both SHOP and WAIMAI are publicly available without licence requirements. There is a dataset user agreement for YELP dataset, which disallows commercial usage of the datasets but encourages academic study. Same thing applies to the IMDB dataset.

For facial emotion recognition, we use four datasets: FER+, RAFDB, EXPW, and AFNET. All the datasets are annotated by the standard seven basic emotions, i.e., {anger, disgust, fear, happy, sad, surprise, neutral}. The images in FER+ (Goodfellow et al., 2015) are from the ICML 2013 Workshop on Challenges in Representation. We use the provided testing portion in FER+. RAFDB (Li et al.) and AFFECTNET (Mollahosseini et al., 2019) were annotated with both basic emotions and fine-grained labels. In this paper, we only use basic emotions since commercial APIs cannot work for compound emotions. EXPW (Zhang et al.) contains raw images and bound boxes pointing out the face locations. Here we use the true bounding box associated with the dataset to create aligned faces first, and only pick the images that are faces with confidence larger than 0.6. We cotnacted the creators of RAFDB and AFNET to obtain the data access for academic purposes. FER+ and EXPW are both publicly available online without consent or licence requirements.

For spoken command recognition, we use DIGIT, AMNIST, CMD, and FLUENT. DIGIT (Dat, d) and AMNIST (Becker et al., 2018) are spoken digit datasets, where the label is is a spoken digit (i.e., 0-9). The sampling rate is 8 kHz for DIGIT and 48 kHz for AMNIST. Each sample in CMD (Warden, 2018) is a spoken command such as "go", "left", "right", "up", and "down", with a sampling rate of 16 kHz. In total, there are 30 commands and a few white noise utterances. FLUENT (Lugosch et al.) is another recently developed dataset for speech command. The commands in FLUENT are typically

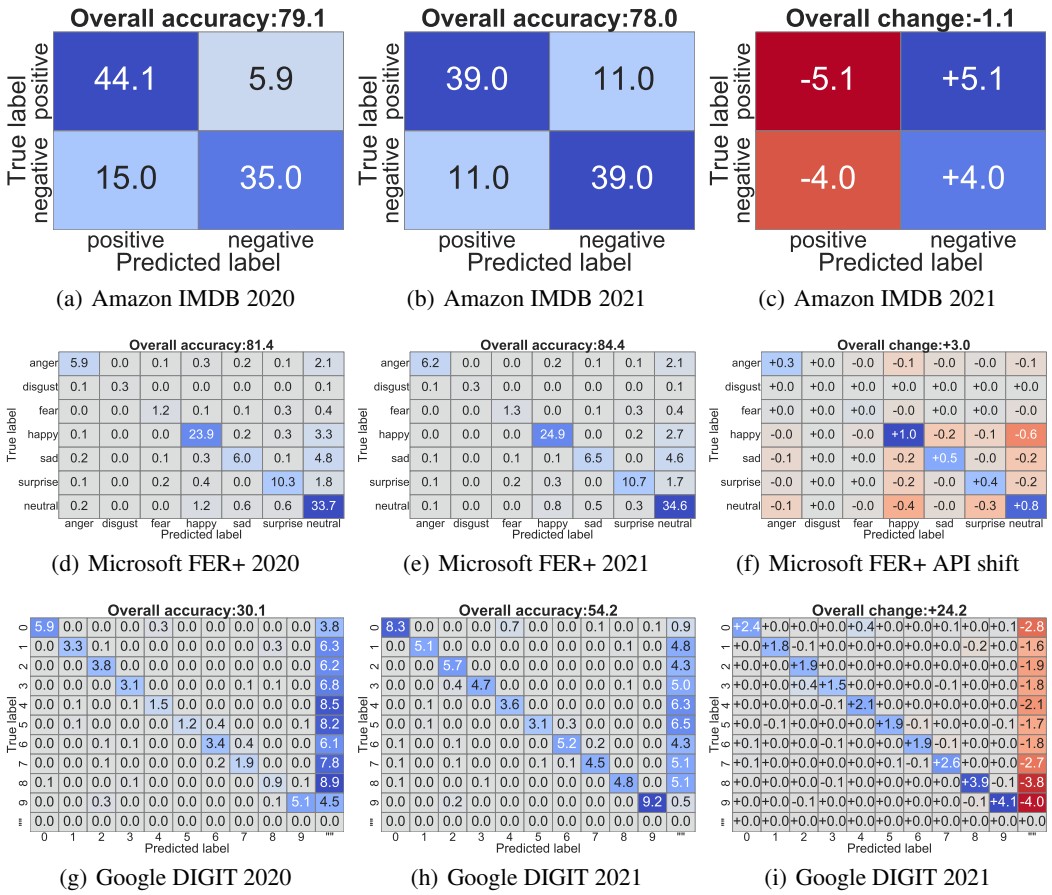

Figure 6: Confusion matrices of a few APIs in spring 2020/2021, along with their API shifts.

a phrase (e.g., "turn on the light" or "turn down the music"). There are in total 248 possible phrases, which are mapped to 31 unique labels. The sampling rate is also 16 kHz. All those datasets are freely available online for academic purposes.

Some of the datasets may contain personal information. For example, the human faces contained in the facial emotion recognition dataset may be deemed as personal information. On the other hand, our study focuses on whether there is a performance change on the dataset, and does not use or disclose any personal information.

For sentiment analysis, we use the Google NLP API (GoN), Amazon Comprehend API (Ama), and the Baidu NLP API (Bai). For facial emotion recognition, we use Google Vision API (Goo, a), Microsoft Face API (Mic, a), and the Face++ API (Fac). For spoken command recognition, we adopt Google speech API (Goo, b), Microsoft Speech API (Mic, b), and IBM speech API (IBM).

**Details of observed ML API Shifts.** Now we present a few more observed ML API shifts, as shown in Figure 6. One observation is that individual entry's change in the API shift can be larger than the overall accuracy's. For example, as shown in Figure 6 (c), the overall accuracy change is about -1.1% for Amazon on IDMB, but the performance drop for positive texts is as large as 5%. This indicates the importance of using fine-grained confusion matrix difference to measure API shifts. In addition, when the overall accuracy increases, it is possible that the accuracy for each label has been improved. This can be easily verified by Figure 6 (d-f). On the other hand, as shown in Figure 6 (g-i), Google API's large accuracy improvement (24%) is mostly because it is able to correctly predict many samples that were previously deemed as empty. One possible explanation is that Google API internally uses a higher threshold to generate a recognition. When the number of label increases, it

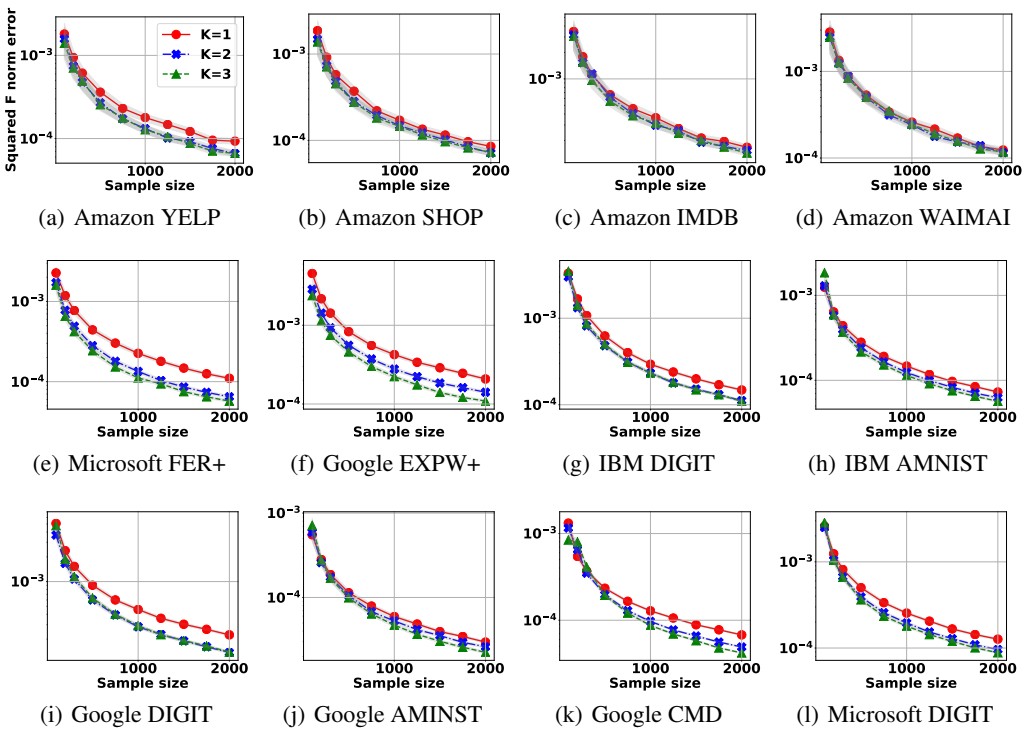

(a) Amazon YELP     (b) Amazon SHOP     (c) Amazon IMDB     (d) Amazon WAIMAI

(e) Microsoft FER+     (f) Google EXPW+     (g) IBM DIGIT     (h) IBM AMNIST

(i) Google DIGIT     (j) Google AMINST     (k) Google CMD     (l) Microsoft DIGIT

Figure 7: Effects of partition parameter $K$. The total number of partitions is $LK$, and thus Larger $K$ implies more partitions. Generally, across 12 cases where API shifts are identified, larger number of partitions usually leads to smaller estimation error for large samples. In practice, we observe that $K = 3$ is enough to reach good error rate.

might become hard to manually check the API shifts. For those cases, an anomaly detector can be applied to quickly identify the most surprising components in the API shifts.

**Partition size's effects on MASA.** Now we study how the partition number affects the performance of MASA, as shown in Figure 7. Across all API shifts we estimated, we note that larger number of partitions leads to a smaller overall Frobenious norm in general. This is expected, as larger $K$ effectively introduces more parameters to estimate and thus is more powerful. The trade-off is that the computational cost increases, and more samples are needed for initial estimation. Interestingly, as $K$ becomes large, the relative error reduction improvement becomes small. This is probably because there is no strong uncertainty difference within small partitions. In practice, we found that $K = 3$ already gives a small enough error reduction.

**Comparison with baselines for case study on YELP.** To further understand MASA's performance, We compared the performance of MASA with two baselines: random sampling and standard stratified sampling (proportionate allocation). We drew 2000 samples for all methods, and repeated the experiments 1000 times to obtain an average of the Frobenius norm error. MASA outperforms both baselines significantly: the observed error is 0.015 for random sampling, 0.009 for stratified sampling, and 0.006 for MASA.

**Understanding uncertainty score.** MASA is developed based on the notion of uncertainty scores, and thus it is worthy understanding how uncertainty scores of different partitions for an ML API are computed. Here, we provide an illustrative example, as shown in Figure 8. The dataset contains three partitions and each partition includes six data points. We use a small ball to represent each data point, its interior color to denote its true label, and its edge color to denote the predicted label of an evaluated ML API. For example, on partition 1 and partition 3, all edge colors match interior colors,

| Partition 1 | Partition 2 | Partition 3 |
|:-:|:-:|:-:|
| 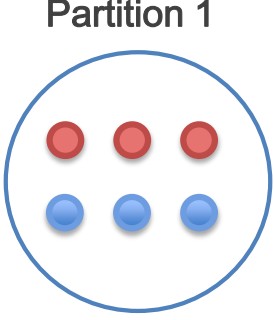 | 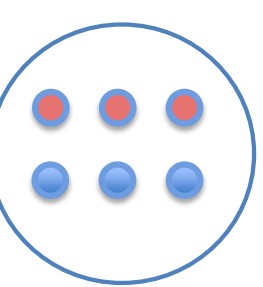 | 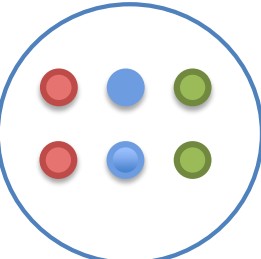 |
| Accuracy: 1.0 Uncertainty: 0.50 | Accuracy: 0.5 Uncertainty: 0.00 | Accuracy: 1.0 Uncertainty: 0.67 |

Figure 8: Illustrative examples of uncertainty scores. The dataset contains three partitions, each of which includes six data points. Here we use a ball to represent a data point, its interior color to denote its true label, and its edge color to indicate an ML API's predicted label. For example, as shown in the left panel, three points are labeled as red and the other three are labeled as blue. All points are predicted correctly, and thus the accuracy is 1.0. As the ML API predicts half of the points as red and half as blue, the uncertainty score is $1 - 0.5 \times 0.5 - 0.5 \times 0.5 = 0.5$. Note that high accuracy does not necessarily imply low uncertainty. For example, accuracy on partition 1 (1.0) is higher than that on partition 2(0.5), but its uncertainty score is actually larger than the latter. Yet, high diversity in the predicted labels does imply higher uncertainty. For example, while accuracy on partition 1 and partition 3 are both perfect, partition 3 incurs a higher uncertainty. This is because while only two unique predicted labels exist in partition 1, three occur in partition 3.

and thus the accuracy is 1.0. On partition 2, interior and edge colors match only on half of the points, and thus the accuracy is only 0.5.

To understand the calculation of the uncertainty score, let us take partition 1 as an example. The ML API predicts the label red for half of the partition and blue for the other half. Thus, the uncertainty score is 1 subtracting the sum of the square of likelihood of each predicted label, i.e., $1 - 0.5 \times 0.5 - 0.5 \times 0.5 = 0.5$. Similarly, on partition 2, the ML API always predicts the label blue, and thus the uncertainty is simply $1 - 1 = 0$. On partition 3, the ML API evenly predicts three unique labels, and thus the uncertainty score becomes $1 - \frac{1}{3} \times \frac{1}{3} - \frac{1}{3} \times \frac{1}{3} - \frac{1}{3} \times \frac{1}{3} = \frac{1}{3} \approx 0.67$.

Two observations are worthy mentioning about uncertainty scores, in addition to their non-negativity and upper bound of 1. First, higher accuracy on a partition does not imply lower uncertainty. To see this, note that the accuracy on partition 1 (1.0) is higher than that on partition 2 (0.5), but its uncertainty score is actually larger than that of partition 2. In fact, an API's accuracy on a partition is orthogonal to its uncertainty, as uncertainty score only depends on the predicted labels and is independent of the true labels. Second, diversity of the predicted labels is correlated to the uncertainty score. For example, the accuracy is same on partition 1 and 3, but the uncertainty score is higher on partition 3, mainly because there are three unique labels (red, blue, and green) in partition 3. This is expected, as uncertainty scores are designed to capture how diverse an ML API's prediction can be.

