# OpenReview forum: "How Did the Model Change? Efficiently Assessing Machine Learning API Shifts "
_ICLR.cc/2022/Conference — ICLR 2022 Poster_

### Official Review · Reviewer_2mEB · 2021-10-25

**Correctness:** 3
**Technical Novelty And Significance:** 3
**Empirical Novelty And Significance:** 4
**Recommendation:** 6
**Confidence:** 4

**Main Review:**

Strengths
- The use case is one that is not often studied, yet is important both from a business accountability perspective (assessing the trustworthiness of a commercial ML API) and ethics perspective (providing information to society about hidden dangers in commercial ML APIs, which are difficult to track because of the algorithmic intransparency that is a common problem in the tech industry)
- Algorithm is uncomplicated to implement, and provides clear improvement over naive sampling methods
- Collected dataset on API shifts is novel, and is a major contribution of the work if released to the public

Weaknesses
- The idea of difficulty levels (K) is not well-developed. There are a couple of places where I was expecting more details: (1) some experiments use K=2 while others use K=3, what is the justification for doing so?; (2) I was expecting an ablation study comparing MASA with K=1 vs MASA with K>1. As the paper stands, it is unclear just how much of MASA's performance improvement over uniform and stratified sampling is due to K, versus the uncertainty-based sample selection.

Other suggestions for improvement
- Being able to measure the uncertainty in each partition, and use that uncertainty to inform sample selection, is the key idea that makes MASA perform better than uniform or stratified sampling. Given the importance of this idea, it would be better to explain this via an intuitive figure. As a suggestion, the authors could visualize how a partition in which the ML API classifies all data points with the same class (whether right or wrong) will have low uncertainty, while the opposite gives high uncertainty.

**Summary Of The Paper:**

A method to estimate the confusion matrix of a black-box classifier, using as few samples as possible. The paper is focused around one use case: tracking changes in ML APIs, considering that such changes may go unannounced.

**Summary Of The Review:**

The paper makes atypical but important contributions to ML ethics. Although the proposed algorithm is not groundbreaking from a technical perspective, it does contribute significantly towards measuring and tracking changes in black-box APIs, and I think it is of high value to society. I am concerned that the idea of difficulty levels (K) is not fully developed, but I lean towards acceptance.

---

> ### Author Response · Authors · 2021-11-18
> **Thank you for your helpful summary and support**
>
> Thank you for your helpful summary and support for the paper! We answer your questions as follows.
>
>
> ***[How to choose the difficulty levels (K) and where is the detailed study?]***: (1) Our case study used K=2 for demonstration purposes only, and K=3 was used for other experiments. This is because empirically we observed that K=3 is sufficient to reach a small error, as shown in Figure 7 (page 31 in the appendix). (2) The ablation study on the effects of K was actually given in Figure 7, page 31, in the appendix. Both K and uncertainty sample selection contribute to the performance improvement.
>
> ***[Visualize the uncertainty-based idea in an intuitive figure]***: We added one figure (Figure 8) in the appendix to visualize the idea and also included a discussion on it.

---

> > ### Comment · Reviewer_2mEB · 2021-11-24
> > **Thank you for the responses**
> >
> > The authors' responses have answered my questions. If the paper is accepted, I still think it would be helpful to (briefly) explain how K was selected in the main paper, or at least refer the readers to the appendix.

---

### Official Review · Reviewer_Z5qk · 2021-11-02

**Correctness:** 4
**Technical Novelty And Significance:** 3
**Empirical Novelty And Significance:** 3
**Recommendation:** 8
**Confidence:** 4

**Main Review:**

Strengths:
- Tackles important practical problem of the result differences from ML APIs
- Presents accuracy change results from a number of actual ML APIs from leading providers
- Authors created a novel algorithm (MASA) to efficiently detect and evaluate result differences for ML APIs.
- Authors demonstrate that MASA significantly outperforms baseline algorithms.

Weaknesses:
- Accuracy changes from actual ML APIs is limited in scope. Only few systems were analyzed and only for two dates (spring 2020 and spring 2021). It would be interesting and important to track both more systems and more time points
- Unclear how confusion matrixes were used for speech recognition task, which presumably has a very large number of "classes". I am guessing authors treated speech recognition problem as classification problem for evaluation. However, there are no details on this. It would be good if this was explained and info provided.
- As authors noted, confusion matrix difference is a good measure as a result drift only for certain (classification-like) APIs. It would be good to see how to deal with non-classification APIs.
- Authors suggest that the differences seen in confusion matrix provide useful insights into how API results changed and why they changed. There is little substantiation of usefulness of how the API results changed. I.e. is confusion matrix difference really the best we can do to show how the API results changed? Regarding "why" the results changed, the authors provide guesses, but it seems to me that we cannot really know based on the results. If we cannot determine the "why", this should be stated. If we can, then it would be good to see what can be determined and how.

Minor typo:
Page 7: diffident - should be different.

**Summary Of The Paper:**

Authors show that ML models behind publicly available APIs change and these changes cause result changes for input datasets.
Authors track the changes through confusion matrix differences. They propose an efficient algorithm they call MASA to evaluate the changes in results with reduced number of queries. Their algorithm achieves better estimates given the same budget than uniform sampling.

**Summary Of The Review:**

I think that the problem of ML API result shift is real and important. I believe authors made interesting and useful contribution in evaluating such shifts. Although the paper has some weaknesses, I would recommend accepting it.

---

> ### Author Response · Authors · 2021-11-18
> **Thank you for your helpful summary and support**
>
> Thank you for your helpful summary and support for the paper! We answer your questions as follows.
>
> ***[Can you track both more systems and more time periods?]***: Yes, we will keep tracking more ML APIs more frequently.
>
> ***[Clarification of labels of speech recognition tasks]***: The original API annotations were adopted from [a1] (also cited in the main paper) and thus we follow their labeling approach. More specifically, for DIGIT and AMNIST, the labels are 0,1,2,...,9. CMD and FLUENT are spoken command datasets and contain 31 unique labels (such as “go”, “left”, “right”, and “up”). The details are also provided in the appendix (page 29).
>
> [a1] L. Chen et al, FrugalML: How to use ML prediction APIs more accurately and cheaply, NeurIPS 2020.
>
>
> ***[Generalization of MASA to ML tasks beyond classification]***: As the first step towards studying the API shifts, this paper is focused on classification. This is still quite useful because most of the ML APIs are classification APIs. We agree with you that extending MASA to more complicated tasks such as segmentation or clustering is an interesting future direction.
>
>
> ***[How do differences in the confusion matrix help explain API shifts?]***: The confusion matrix differences indicate (i) that the APIs have been updated and (ii) how APIs’ update changes their likelihoods of misclassifying one particular label into another. (i) is because the confusion matrix is determined on the same dataset and a difference means that an API makes a different prediction on the same data point. (ii) holds as the confusion matrix difference can be viewed as a decomposition of the accuracy changes, and the trace of the confusion matrix equals the accuracy change. Thus, one can infer which label-to-label misclassification affects the accuracy change more. For example, as shown in Figure 1 (a) and (b), prediction error on label ”4” increases the most among all labels, and most of it is due to predicting “4” as “5” (1.9% error). This may help the users understand which predictions become less reliable and need more inspection.
>
> ***[Typo]***: We fixed this in the revision.

---

> > ### Comment · Reviewer_Z5qk · 2021-11-29
> > **Thanks for your answers**
> >
> > Thanks to authors for their answers to my questions and clarifications.

---

### Official Review · Reviewer_NY7g · 2021-11-02

**Correctness:** 4
**Technical Novelty And Significance:** 3
**Empirical Novelty And Significance:** 4
**Recommendation:** 6
**Confidence:** 3

**Main Review:**

The problem addressed is important and the method seems to yield good results in practise in comparison to random sampling.

Once concern is its unclear why this problem cannot leverage some of the existing work on stratified sampling (based on explanation at the end of section 1) with the aim to reduce the variance of the estimator? Could you please elaborate on this. After all, the goal is to estimate elements of the confusion matrix.

Based on section 3.3, does MASA yield an optimal allocation of API calls?

**Summary Of The Paper:**

API shifts are common in several deployed machine learning models. This work proposes an efficient way to measure shifts in the confusion matrices of ML models using limited number of API calls.

**Summary Of The Review:**

API shift is an important problem that needs to be addressed in order to operationalising AI. This work proposes a way to assess these shifts using limited number of API calls and has a lot of practical importance. Most deployed ML models are also priced in a manner wherein each API calls has an associated cost and thus performing API shift assessment using limited calls is quite important even from a pricing stand point.

---

> ### Author Response · Authors · 2021-11-18
> **Thank you for your helpful summary and support!**
>
> Thank you for your helpful summary and support for the paper! We answer your questions below.
>
> ***[Can stratified sampling be leveraged to minimize variance of the estimator?]***: Standard stratified sampling may lead to lower error than random sampling, but it is unclear whether/how it can be adapted to obtain a near-optimal sample allocation. This is primarily because the optimal sample allocation for confusion matrix estimation relies on the unknown uncertainty scores instead of variance. This makes allocation achieved by stratified sampling sub-optimal and also leaves room for improving sample and computational efficiency. We have revised the discussion in Section 1 to make this clear.
>
>
> ***[Does MASA yield an optimal allocation of API calls?]***: Yes. As shown in Theorem 2, MASA’s performance is asymptotically close to the optimal for any given sample budget N. Empirically, as shown in Figure 4 (g), the empirical sample allocation (solid lines) is almost the same as the optimal allocation (dark dot points).

---

### Official Review · Reviewer_Bj8e · 2021-11-02

**Correctness:** 4
**Technical Novelty And Significance:** 3
**Empirical Novelty And Significance:** 3
**Recommendation:** 6
**Confidence:** 4

**Main Review:**

**Originality and significance**: To the best of my knowledge, this is the first work to systematically investigate the shift in the performance of commercial ML APIs. With the popularity of these services on the rise, it is important to study various aspects of the models, including the variations of the performance over time. The proposed sampling method is also novel and can be used in similar settings where model queries are expensive. The paper aims to minimize the Frobenius norm of the error in estimating the confusion matrix of classifiers, while keeping the sample complexity close to the optimal allocation strategy (in hindsight). The proposed algorithm asymptotically approaches the optimal allocation decay rate of 1/N. The empirical studies suggest that the proposed method can be an order of magnitude more sample efficient compared to random sampling.

**Quality and clarity**: The paper is well written and motivates the problem with case studies and examples. The algorithms and theorems are clearly stated. I did not check the proofs in the appendix.

Some limitations of the work:
- The method (as described in the paper) only applies to classification settings.
- Although not a strict requirement, the experimental analysis of the sampling method benefits from a rough estimate of the “difficulty” of each example before it is evaluated by the ML API . This “difficulty” is calculated by using a separate (client-side) cheap model. This goes against one of the main appeals of ML APIs that try to minimize client-side evaluation setup (think installing TF runtime, etc). There are no ablation studies on the quantified role of using such client side models.
- The experiments could be improved by comparison to baselines other than random sampling. Though this might not be possible with updated APIs.

Note: I was a reviewer to an earlier version of this work. Compared to the previous version, (1) the writing of the paper is improved in parts, (2) there is a more clear discussion of related work, and (3) the dataset is being publicly released.

**Summary Of The Paper:**

This paper considers the problem of estimating the change in the performance of commercial ML APIs (ML as a service) as the models are updated over time (experiments are for 2020 vs 2021). It formalizes the problem as estimating the change in the confusion matrix over time. The main theoretical contribution is an adaptive sampling method to more efficiently estimate this shift. Interesting empirical results on various ML APIs are provided in the paper, showing the relevance of the problem and the effectiveness of the proposed method.

**Summary Of The Review:**

This work opens a discussion around the problem of estimating the performance shift in commercial ML APIs (for classification). The paper defines a metric for the performance shift of such APIs (via the confusion matrix), and presents a method to achieve near optimal sampling rates.

The theoretical contributions of the paper are small but non-trivial. The experimental analysis is detailed and interesting, but could benefit from further ablation studies on the effect of the client-side difficulty gauge model. The problem is of interest to the ML community and the release of the annotated dataset used in this work would be useful to the community.

---

> ### Author Response · Authors · 2021-11-18
> **Thank you for your helpful summary and support!**
>
> Thank you for your helpful summary and support for the paper! We answer your questions below.
>
>
> ***[Generalization to ML tasks beyond classification]***: As the first step towards studying the API shifts, this paper is focused on classification. This is still quite useful because most of the ML APIs are classification APIs. We agree with you that extending MASA to more complicated tasks such as segmentation or clustering is an interesting future direction.
>
> ***[Necessity of difficulty estimator]***: We want to clarify that difficulty scores are not necessarily needed by MASA. Any metadata that correlates with model accuracy can be used in MASA, e.g., image brightness and paragraph length. In fact, MASA can still obtain an improvement without any metadata. For example, we observed a 70% sample size reduction to evaluate Amazon API on YELP given by MASA using labels alone (using the labels alone corresponds to partition number K=1 in Figure 7 in the appendix). While larger K gives a better error-sample trade-off, the performance of K=1 is still much better than standard uniform sampling.
>
> ***[Compared with baselines other than random sampling]***: We evaluated an additional baseline, standard stratified sampling (proportionate allocation), on the case study dataset YELP. We drawed 2000 samples for all methods, and repeated the experiments 1000 times to obtain an average of the Frobenius norm error. MASA outperforms both baselines significantly: the observed error is 0.015 for random sampling, 0.009 for stratified sampling, and 0.006 for MASA. We have added a discussion of this comparison in Supplementary Section C (page 31) of the revised paper.

---

> > ### Comment · Reviewer_Bj8e · 2021-11-29
> > **Comment**
> >
> > The authors' response answered most of my questions and concerns. The added experiment comparing the method to other baselines is a nice improvement. I recommend adding the discussion around *[Necessity of difficulty estimator]* to the main body of the paper (referencing the results in the appendix).

---

### Author Response · Authors · 2021-11-17
**Thank you for the reviews**

We thank all the reviewers for their helpful feedback and support of the paper. We have uploaded a revised manuscript based on your suggestions.

---

### Decision · Program_Chairs · 2022-01-20

**Decision:**

Accept (Poster)

**Comment:**

The paper studies real world ML APIs' performance shifts due to API updates/retraining and proposes a framework to efficiently estimate those shifts.  The problem is very important and the presented approach definitely novel. My concern is about limited novelty of the theoretical analysis and weak experimental evaluation (just two dates, limited number of systems tested, small number of ablations). As of now the paper looks like an interesting but unfinished proposal. Looking forward to the discussion between the authors and the reviewers to address the concerns.

In the rebuttal, the authors have addressed reviewers' comments, in particular by adding additional experiments that strengthen the paper. All the reviewers recommend the paper to be accepted. It is suggested that in the camera-ready version the authors will add additional details regarding the experiments, as some of the reviewers mentioned.